# Hydrogenation of saturated organic and inorganic molecules in metallic hydrogen

Jakkapat Seeyangnok [1,2], Udomsilp Pinsook [1] ✉ & Graeme J. Ackland [2] ✉

Metallic hydrogen is the most common condensed material in the universe, however, experimental studies are extremely challenging, and understanding of this material has been led by theory. Chemistry in this environment has not been probed experimentally, so here we examine carbon, nitrogen, and oxygen in metallic hydrogen using density functional theory calculations. We find that carbon, nitrogen and oxygen react with each other and metallic hydrogen to produce molecules with covalent-type bonding based on sixfold coordinated carbon, threefold oxygen and fourfold nitrogen: $CH_6$, $C_2H_8$, $C_3H_{10}$, $OH_3$, $NH_4$, and $CH_4OH$. In view of the excess hydrogen we refer to them as hypermethane, hyperethane etc. This work suggests that molecular chemistry may take place in very different environments from those found on earth, and may be common throughout the universe. Furthermore, the solubility of C, O, and N casts doubt on whether rocky cores can exist in giant planets.

Metallic hydrogen[1] is believed to be the most common condensed phase of matter in the universe, comprising the cores of gas-giant planets[2–4] and giving rise to their enormous magnetic fields. However, it is exceptionally challenging to make metallic hydrogen on Earth[5–8]. Consequently, most of our understanding of this material comes from theory[9,10]. Most elements become metallic at sufficiently high pressures and, in general, metallic liquids will mix, lowering the melt point. Here, we investigate the solubility of other light elements in a metallic hydrogen environment.

The chemical bonding in these systems can be described in detail using electronic structure calculations, however the coexistence of localised and delocalized electrons means that descriptions used in conventional chemistry must either be discarded or extended. In this analysis, "covalent" bonds will be defined in terms of Crystal Orbital Hamilton Population (COHP) analysis.

Planetary models depict layers separated by the weight of elements, e.g. gas giants feature an outer molecular hydrogen and helium envelope and a core of metallic hydrogen depleted of helium[4,11–13] which is predicted to be insoluble below its own metallization conditions[14–21]. Understanding material properties allows us to infer the composition and structure of exoplanets from their mass-radius relation[2,3,14,22–26].

Chemical bonding is different at high pressure. For example, on Earth, the major components of the mantle exhibit sixfold coordinated silicon[27–31], in contrast to the fourfold sp³ bonding found in normal conditions. The unconventional formation of sixfold coordinated silicates are well described by density functional theory calculations as being enabled by the electrons donated from positive ions, and lower mantle minerals are stabilised at pressure thanks to the increased density[32–38]. If chemistry can change so radically just few thousand kilometres beneath our feet, how different might it be elsewhere in the solar system? Network formation and fourfold-sixfold coordination change in carbon has been subject to much discussion based on experimental and theoretical work[39–46].

While helium in hydrogen, and hydrogen-rich metals are very well studied at high-pressure, particularly with potential applications to superconductivity, less attention has been paid to the issue of solubility of heavier elements in metallic hydrogen, and the implications for different chemical bonding within giant planets[21,47–49].

Theoretical study of metallic hydrogen began in 1935, when Wigner and Huntington[1] used free electron theory to estimate the density of metallic hydrogen, obtaining a value remarkably close to current estimates. This implied that hydrogen molecules would transform into atomic metallic hydrogen when subjected to

[1]Department of Physics, Faculty of Science, Chulalongkorn University, Bangkok, Thailand. [2]Centre for Science at Extreme Conditions, School of Physics and Astronomy, University of Edinburgh, Edinburgh, United Kingdom. ✉e-mail: udomsilp.p@chula.ac.th; gjackland@ed.ac.uk

sufficiently high pressure - unfortunately, their estimate of 25 GPa was an order of magnitude too low. In 1968, Ashcroft made a remarkable prediction that metallic hydrogen would be a room-temperature superconductor although currently, theory indicates that at low temperature hydrogen remains molecular to about 500 GPa[9,50–55]. Predictions of the crystal structure of metallic hydrogen came even later[9,50,56,57], through ab initio random-structure searches. Surprisingly, the atomic and metallic crystal phase is now believed to be a complex, open structure, rather than the dense-packed structures assumed by Wigner, Huntingdon and Ashcroft. This type of open structure is typical of the high-pressure Group I electride materials, having $I4_1/amd$ symmetry, isostructural with cesium IV[58–61]. This $I4_1/amd$ structure persists until 2.5 TPa, beyond which more densely packed structures are favoured[50,62]. Fluid metallic hydrogen is expected to exist at much lower pressures, because the covalent bond is broken not only by pressure but also by higher temperatures[24,63–66]. Indeed, since fluid atomic hydrogen is supercritical, this phase may extend to the ultra-low pressure of the interstellar medium[67].

In experiments[68], synthesis of solid metallic hydrogen has been claimed at pressures exceeding 420 GPa using infrared absorption measurements[5,6], and at an even higher pressure of 495 GPa as evidenced by reflectivity measurements[7]. Liquid metallic hydrogen has indeed been reported at much lower pressures in both dynamic and static experiments[8,69,70]

Carbon is particularly important, being the fourth most abundant element, and the building block of organic chemistry. Determining the properties of these materials is essential for understanding the interior of giant planets, such as Neptune and Uranus[71–74], and Jupiter and Saturn[75,77]. Hydrocarbons have recently been studied at planetary interior conditions, i.e. high pressure and high temperature[49,71–87]. Some molecular compounds, such as $CH_4(H_2)_2$[82,83], $C_2H_6$ and $C_4H_{10}$[74], can be formed under these extreme conditions. In addition, there were a number of theoretical studies on helium, methane and various metals in the metallic hydrogen environment[49,75,88]. Studies on giant planets suggest that hydrocarbons may exist within their non-metallic middle layers[73,74,86]. Methane ($CH_4$) has been identified as the most abundant hydrocarbon at pressures of up to several hundred GPa and forms hydrogen-rich compounds with $H_2$ up to 160 GPa[82,89]. At higher pressures and temperatures, simulations and experiments have suggested that methane decomposes into hydrogen and diamond[49,73,86,90]. Due to its density, the latter subsequently gravitationally sinks deeper into the planet in a phenomenon known as diamond rain[73,86,91–99]. This predicted demixing contrasts with the observation of high-pressure reaction between diamond and hydrogen[100] - an issue which has caused significant practical challenges to synthesizing metallic hydrogen in diamond anvil cells[68].

One challenge for theory is the richness of hydrocarbon chemistry. The demonstration that methane is unstable to decomposition does not preclude other stable hydrocarbons. Moreover, given the high temperatures and excess of hydrogen over carbon in gas giant planets, even a low solubility limit could result in much of the carbon remaining in solution in the metallic hydrogen layers.

Here, we use density functional theory calculations to consider what form of carbon will exist in a metallic hydrogen environment. We start by examining the free energy of solid solution carbon in crystalline $I4_1/amd$ metallic hydrogen with the well-characterized quasi-harmonic method. Then we demonstrate the equivalence of the molecular dynamics (MD) approach as an equally reliable estimator of thermodynamic properties, and apply MD to investigate the planetary-relevant fluid metallic hydrogen.

Our result is that we predict the existence of a chemistry, based around a basic sixfold coordination of carbon, fourfold coordination of nitrogen and threefold coordination of oxygen. For example we observe $CH_6$, $C_2H_8$, $C_3H_{10}$, $OH_3$ and $CH_4OH$ and $NH_4$. These hyper-molecules become stable over methane, water and ammonia in conditions of high pressure and modest temperatures, in both solid and liquid metallic hydrogen.

It is worth noting that sixfold coordinated carbon has previously been synthesized[101,102] in $C_6(CH_3)_6^{2+}$ and in the nitrogenase iron-molybdenum cofactor[103]. The 2+ ions are described as non-classical carbocations, with multicentred covalent bonds. However, our hypermolecules only exist in a metallic environment, and we will show that the molecules are stabilised by the high electron density rather than the pressure per se.

## Results

### Solid solubility of carbon in $I4_1/amd$ metallic hydrogen

The solid solubility of carbon in metallic hydrogen can be calculated using the Gibbs free energies (Eq. (1)). Hydrogen exhibits strong nuclear quantum effects[66,104], so our calculation includes the zero-point energy of the system as well as the enthalpy, configurational entropy, and vibrational entropy.

$$G(P,T) = H(P,T) + U_{ZPE} + TS_{con} + TS_{vib} \tag{1}$$

Initially, we consider a single carbon solute in hydrogen (Supplementary Fig. 1). This is more complicated than the standard solid solubility calculation[105–107] because carbon is significantly larger than hydrogen, so substituting one carbon atom for a single hydrogen is not the most stable arrangement. We tried removing clusters of up to eight hydrogens and found that the most stable arrangement involves removing five hydrogens (Table 1, Fig. 1). If more than five hydrogens are removed from the starting configuration, a vacancy defect is created which diffuses through the lattice in MD.

We used the quasiharmonic approximation, and as a check, particularly of anharmonic effects, we also calculated the free energy from the phonon density of states derived from the velocity autocorrelation function of an MD calculation. The results agree very well for carbon and reasonably well for hydrogen: this is expected from the different anharmonicity in the two systems which is better captured by the MD.

From both MD and static enthalpy, it shows that the classical enthalpy prefers pure substances to the mixture with positive values $\Delta H_{MD}$ and $\Delta H_{Static}$. On the other hand, the entropy and zero-point energy[108–110], favour the mixture, with negative values in both $-T\Delta S$ and $\Delta U_{ZPE}$ (see Table 1 and Fig. 1a). For the most favourable arrangement, the free energy has an impurity formation energy of $\Delta g = 0.39 \pm 0.17$ eV. We always observe an octahedral arrangement of six neighbouring hydrogens, forming $CH_6$.

The overall electronic density of states (Fig. 1) is uninformative, being dominated by characteristic free-electron form, with the hydrogen atoms arranged in a way to produce a pseudogap at the Fermi energy, similar to isostructural Cs-IV[59,111]. However, the electron localization function (ELF)[112] around the carbon (Fig. 1d–f) shows that

### Table 1 | Thermodynamic properties for the solid solution at 300 K and 500 GPa

| Supercell | $\Delta H_{MD}$ | $\Delta H_{static}$ | $-T\Delta S$ (MD) | $\Delta U_{ZPE}$ (MD) | $g_{sol-MD}$ |
|---|---|---|---|---|---|
| $CH_{126}$ | $3.12 \pm 0.13$ | 2.926 | $-0.635$ | $-0.798$ | $1.69 \pm 0.13$ |
| $CH_{125}$ | $2.90 \pm 0.18$ | 2.667 | $-0.582$ | $-1.376$ | $0.94 \pm 0.18$ |
| $CH_{124}$ | $2.51 \pm 0.14$ | 2.587 | $-0.484$ | $-1.328$ | $0.69 \pm 0.14$ |
| $CH_{123}$ | $3.10 \pm 0.17$ | 2.681 | $-0.752$ | $-1.957$ | $0.39 \pm 0.17$ |

Calculated energies are relative to the fully ionised atomic states. $H_{MD}$ represents the ensemble average enthalpy obtained from NPT MD at 500 GPa, while $H_{static}$ is derived from static optimization. Entropy $TS$ is the summation of both vibrational and configurational entropy ($S_{vib} + S_{conf}$). $U_{ZPE}$ and $S_{vib}$ were calculated using the phonon density of states. All energy terms are compared with an unmixed reference state, quoted in units of eV per carbon atom. The calculated enthalpy $H$, entropy $TS$, zero-point energy $U_{ZPE}$, and configurational entropy, Gibbs free energy are given in Supplementary Table 1.

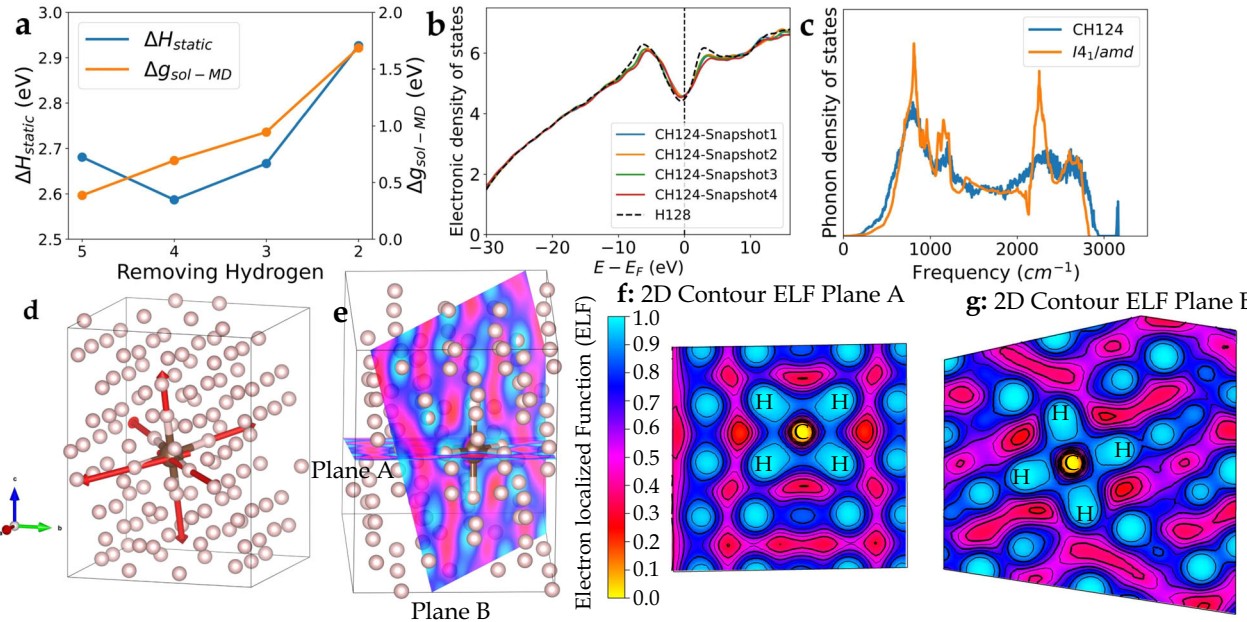

**Fig. 1 | Enthalpy and Gibbs free energy differences, electronic density of states, and electron localised function (ELF) of carbon in solid, metallic hydrogen.** **a** Enthalpy of solution (eV/atom) as a function of number of removed hydrogens from static relaxation (blue) and NPT-ensemble MD (orange). **b** Electronic density of states from a typical BOMD snapshot, showing the characteristic free electron form with a pseudogap at the Fermi Energy. **c** Phonon density of states from DFPT of a structure from a relaxed $CH_{124}$ MD snapshot (blue), compared with pure $I4_1/$ *amd* hydrogen (orange). **d** Visualization of eigenvectors of the characteristic symmetric stretch mode of $CH_6$ as shown in the sharp peak above 3000 cm$^{-1}$ in part (**c**). **e–g** Electron Localization Function (ELF) surrounding a single carbon atom, including cross-sections of planes A and B. The high ELF values suggest the presence of covalent-type bonds, rather than the metallic bonding elsewhere. The zero value for ELF on the carbon atom is an artefact arising from the 1s electron being described by the pseudopotential.

electrons are localized between the hydrogens and carbon in $CH_6$, but not between hydrogen pairs (Supplementary Fig. 2). This suggests that carbon has reacted to form a $CH_6$ molecule with six covalent CH bonds, which we call hypermethane (Supplementary Fig. 3 and Supplementary Table 2).

The phonon calculations show that pure hydrogen exhibits the expected two-peaked acoustic and optical branches of $I4_1/amd$. The $CH_{124}$ supercell retains smeared-out versions of these modes, and the heavier carbon reduces the frequency of the acoustic modes in the supercell. The $CH_6$ molecule has a well-defined symmetric stretch mode, at a frequency higher than anything in the pure metallic hydrogen. (Fig. 1c, d), We compute this mode at the $\Gamma$ point for four optimized structures from four different snapshots. We found that all of our calculations show the distinctive peak at $3161 \pm 2$ cm$^{-1}$. Other modes involving asymmetric CH stretches are mixed with the highest frequency $I4_1/amd$ modes.

The solid solution gives us an indication that, like silicon on Earth, carbon will go from fourfold to sixfold coordination in giant planets. However, the positive heat of solution suggests that temperatures well above the melting point would be needed for significant solubility. We investigate this further in the next section.

## Hypermethane in liquid metallic hydrogen

To further investigate this hypermethane, we simulate a $CH_6+H_{118}$ supercell in the NPT ensemble at 500 GPa. The radial distribution function (RDF), (Fig. 2a), indicates that at 300 K, we have well-defined H·H peaks representing the $I4_1/amd$ crystal. In the 600 K and 900 K cases, the structure melted as expected[10], indicated by the smoothness of the RDFs (Fig. 2a). The mean square displacement (MSD) confirms the melting transition, with a stable MSD at 300 K and a linear increase at 600 K and 900 K (Fig. 2b).

In all cases, the carbon-hydrogen RDF (Fig. 2d), has a strong peak between 1.0 Å, and 1.3 Å. Even under melted conditions, we observe

hydrocarbon $CH_6$ hypermolecules (Fig. 2e) where the cumulative number of bonds in the first peak of the RDF is six. Moreover, the angle distributions (Fig. 2c, f) also indicate an octahedral arrangement of six neighbouring hydrogens in both solid and liquid metallic hydrogen. These results suggest the existence of the hypermethane $CH_6$ molecule above the melting line of metallic hydrogen at 500 GPa.

## Other hypermolecules: $CH_6$, $C_2H_8$ $C_3H_{10}$, $H_3O$, $NH_4$ and $CH_4OH$

We now investigate whether more complex molecules can form in liquid metallic hydrogen using NVT MD at around 500 GPa and 600 K. We investigated six cases, adding a single carbon, nitrogen, or oxygen atom, a $C_2$ dimer, $C_3$ trimer, and a CO molecule to liquid metallic hydrogen.

In each case, a spontaneous reaction took place with the metallic hydrogen to produce a well-defined, stable hypermolecule with NH, OH and CH bondlengths oscillating in the range 1.0–1.30 Å. We identify these molecules as hypervalent[113] because they are hydrogen rich: $CH_6$ $C_2H_8$ $C_3H_{10}$, $NH_4$, $H_3O$ and $CH_4OH$. These correspond to sixfold coordination for carbon, fourfold for nitrogen and trivalent oxygen, with CC and CO double bonds (Fig. 3). We note that this mimics the high-pressure behaviour of the equivalent second row elements: hexavalent silicon[32], tetravalent phosphorus, and trivalent sulphur[114].

Figure 3 shows the CH, NH and OH radial and cumulative distribution functions for the hypermolecules we investigated, with the sharp first peak in RDF defining the covalent bond, and the plateau in cumulative distribution function (CDF) showing the coordination. All hypermolecules remain stable throughout the 10ps simulation. The angular distribution of these molecules (Supplementary Fig. 4) also supports these findings, with peaks corresponding maximum-angle geometry such as octahedral $CH_6$ molecules (Fig. 3i, j).

The ensemble-averaged enthalpic part of the solubility $\Delta H$ in the liquid compared to the separate elements (Table 2) is negative, unlike the solid. The very high absolute enthalpy values for O, N compounds

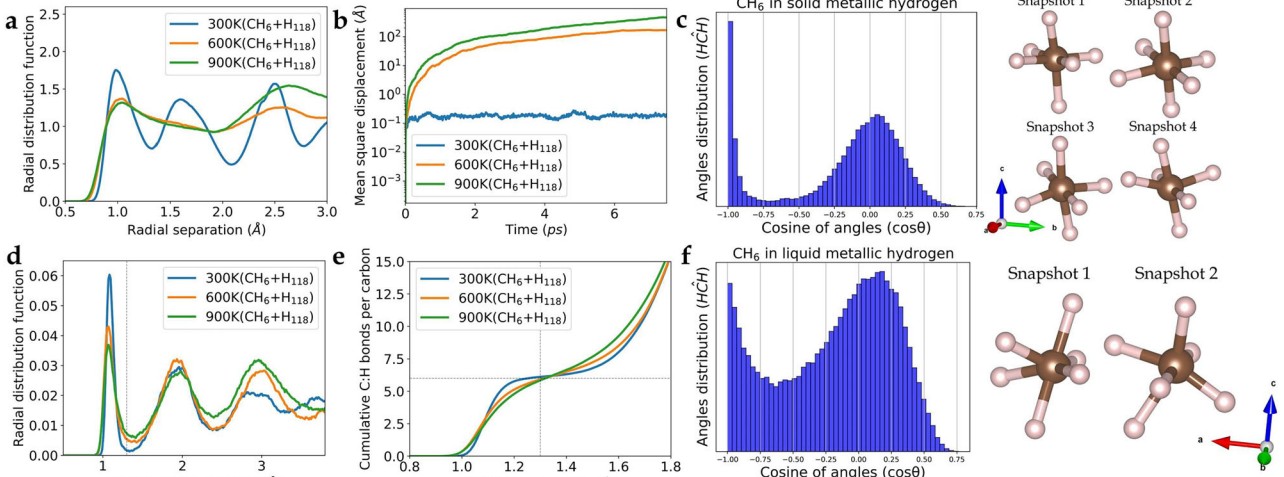

**Fig. 2 | Radial function distribution, mean square displacement, and cumulative number of bonds per carbon from BOMD in NPT ensemble. a** The radial distribution function of hydrogen atom in the $CH_6+H_{118}$ BOMD simulations under the NPT ensemble at varying temperatures, and 500 GPa. The blue solid line denotes the presence of structural peaks in $CH_6+H_{118}$, while the orange and green solid lines signify the disappearance of these peaks due to system melting. **b** Mean square displacement (MSD) of $CH_6+H_{118}$ is examined, with the MSD depicting the crystal structure of $CH_6+H_{118}$ with a finite MSD at 300 K, represented by the blue solid line. Conversely, cases of melting at 600 K and 900K exhibit increased MSD. **c** The angle distribution of $CH_6$ in solid metallic hydrogen at 300 K between pairs of hydrogen atoms with respect to carbon for each step of MD within the radius cutoff of 1.30 Å where the average number of coordination is about 6 hydrogen atoms. Two significant peaks at 90 and 180 degrees demonstrate that the molecule is a octahedron, illustrated by four snapshots of the hypermolecule. **d** The radial distribution function illustrates the distribution of carbon-hydrogen pairs. The first peak of the CH-pair radial distribution function lies between 1.0 Å and 1.3 Å. **e** Cumulative number of bonds (CH bonds per carbon), indicating the count of hydrogens surrounding each carbon atom: the plateau at 6 confirms the hypermolecule. **f** As part (c) at 600 K: the angle distribution of $CH_6$.

are related to choosing the element as the reference state. This energetic stability, combined with dynamic stability in MD, suggests that C, O and N are fully soluble in metallic hydrogen.

## Chemistry of hypermolecules

To investigate the bonding characteristics between the central carbon atom and its six surrounding hydrogen atoms, we conducted a Crystal Orbital Hamilton Population (COHP) analysis[115] on the solids (Supplementary Figs. 5–7). The COHP projected onto the $s$ and $p$ orbitals for the hydrogen and carbon atoms shows a bonding character below the Fermi energy (valence electrons) and unoccupied anti-bonding states above. Notably, the results suggest orbital mixing ("hybridization") between the $s$ and $p$ orbitals, which is similar to the typical hybridization observed in hydrocarbon molecules under ambient conditions. However, the analysis of whether the bonds are fractionally occupied is ambiguous because the bonding states are degenerate with free electron states.

To further investigate the meaning of COHP in this system, we also analysed the COHP between various other atomic pairs (see Supplementary Fig. S9 in the Supplementary Material). The results show a vanishing COHP between the hydrogen atoms in the first shell, indicating no bonding or antibonding characteristics. The COHP between hydrogen atoms in the first and second shells exhibits weak bonding character, with values comparable to those of the COHP between nearest-neighbour hydrogen pairs in the fourfold coordinated $I4_1/amd$ hydrogen structure. Therefore, these results suggest that the C-H bonds exhibit significant bonding and antibonding characteristics, while other pairs are similar to metallic hydrogen bonding.

Any chemical interpretation requires associating electrons with bonds or localised orbitals[116,117]: density functional theory[118] proves that physical ground state properties are independent of such definitions. Hypermolecules exist in a metallic background, making the already-arbitrary counting of electrons particularly challenging. Furthermore, Kohn–Sham eigenstates follow the symmetry of the molecule and so are intrinsically multicentred, even for methane (see SM). Nevertheless, calculations suggest that the hypermolecules are anions rather than cations, with extra electrons acquired from the free electron background.

Sixfold coordinated carbon, a so-called non-classical carbocation, has been observed[101]. It understood through counting electrons in multicentred bonds by analogy to boron[119,120].

A simplified description of the hypermolecules within DFT comes from replacing the metallic hydrogen with a free-electron jellium[121]. This omits the effect of explicit pressure, focussing only on the electronic structure. Such calculations produce very distinctive electron densities of states with a broad free electron band and sharp localised modes on the hypermolecules (Supplementary Fig. 10 and Supplementary Table 3). This provides a quantitative insight into the electronic aspects of the hypermolecule stability, in the absence of pressure from surrounding $H_2$ atoms, or complications of the $H_2$ structure. With no excess electrons, the $CH_6$ molecule is unstable and dissociates into $CH_4$ and $H_2$. As free electrons become available, two are captured into localised modes: octahedral symmetry means these extra electrons would have to partially fill a triplet state, which is degenerate with free electrons near the bottom of the conduction band. In jellium, this leads to a Jahn–Teller distortion: $CH_6$ is then stable, but the $CH_4$ and $H_2$ combination still has lower energy until an electron density is equal to $40e^-/nm^3$, or $90e^-/nm^3$ if the Jahn–Teller distortion is suppressed.

To a first approximation the singlet and triplet molecular orbitals maintain a constant energy below the conduction band, but the additional localised electrons form states which always lie at the bottom of the conduction band, and their energy is reduced as the band broadens.

Finally, any charged particle in a metal will be screened, giving slowly decaying Friedel oscillations[122,123] in the electrostatic potential. Fluid metallic hydrogen is a more complicated case because both protons and electrons play a role in the screening. Our large simulations with around 1000 atoms shows that there are Friedel oscillations in the proton density extending to seven distinctive peaks for $CH_6$. Such long-range structure can best be explained by screening of a hypermolecular charge.

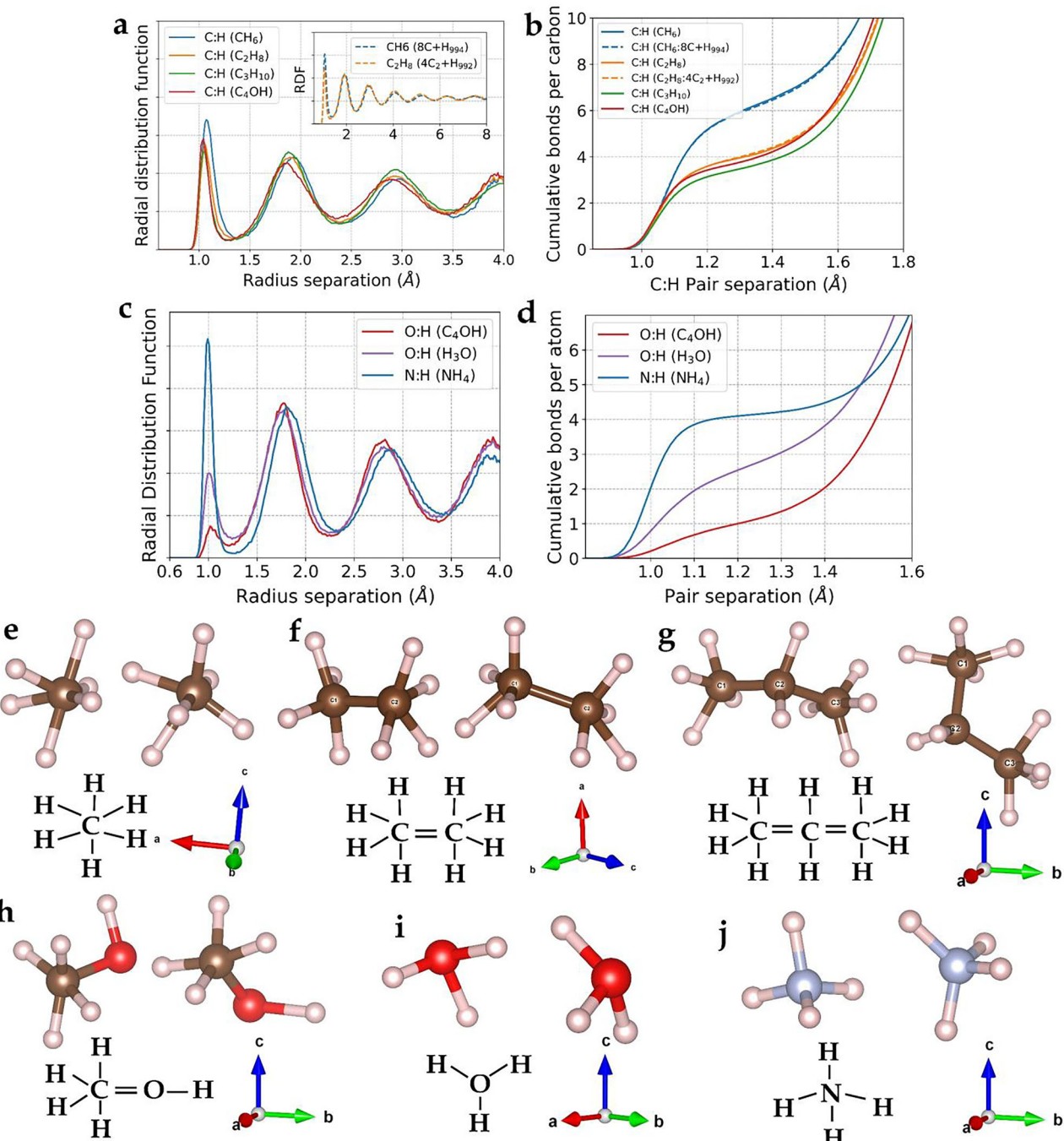

**Fig. 3 | Radial function distribution, cumulative number of bonds in the hypermolecules, and representative snapshots from liquid simulations.**
**a** Displays the RDF of the carbon-hydrogen pairs in metallic hydrogen with one carbon (blue line), two carbons (orange line), three carbons (green line), one oxygen and a CO pair (red line), indicating a smearing of the first peak from 1.0 Å to 1.30 Å. In the ~$10^2$ atom simulations (main figure), structure extends throughout the supercell, however the ~$10^3$ atom simulations (inset) indicates a suppression of these structural oscillations at greater distances. **b** Shows the CDF (number of CH neighbours per carbon) at

1.30 Å, revealing values of ~6, 4, 3.3, and 4, respectively. The short-range order is nearly identical in both system sizes (solid lines: ~$10^2$ atoms, dashed lines ~$10^3$ atoms) (See also in Supplementary Fig. S3 in the supplementary material) (**c**, **d**) equivalent RDF and CDF for OH and NH pairs, showing well-defined and long-lived bonding. These values suggest the formation of molecules, as depicted in (**e**–**i**): Two snapshots from each MD showcasing novel chemical species in liquid metallic hydrogen where the radial cutoff is set at minimum of RDFs of Fig. (**a**) and (**c**), including **e** $CH_6$, **f** $C_2H_8$, **g** $C_3H_{10}$, **h** $CH_4OH$ and **i** $OH_3$ and **j** $NH_4$, with schematic molecular bonding shown below.

## Discussion

In this study, we used density functional theory calculations to predict the existence of hypermolecules such as $CH_6$, $C_2H_8$, $C_3H_{10}$, $OH_3$, $NH_4$ and $CH_4OH$ in metallic hydrogen. These hypermolecules exist only at high pressure in a metallic environment. They are stable both dynamically for the duration of our MD simulations, and energetically against decomposition into elements.

Calculations using jellium in place of metallic hydrogen allow us to determine that the high electron density, rather than the pressure per se, is the key to stabilising the hypermolecules. The hypermolecules can be interpreted as carbon/nitrogen/oxygen forming single bonds to H, and double bonds between heavier elements.

Solubility in the various layers of a planet is a crucial ingredient in planetary modelling.

**Table 2 | The table shows the enthalpy change (ΔH) (eV/atom) of molecular compounds in liquid metallic hydrogen, compared with reference states for carbon, oxygen, and nitrogen, at 500 GPa**

| References | Enthalpy (eV/atom) | Compounds | Enthalpy (eV/atom) | Δ H (eV) |
| --- | --- | --- | --- | --- |
| $H_{128}$ | −9.630(1) | $CH_6$ in $CH_{276}$ | −10.116(1) | −0.16 ± 0.38 |
| $C_{54}$ | −144.087(2) | $C_2H_8$ in $C_2H_{576}$ | −10.096(1) | −0.30 ± 0.60 |
| $O_{48}$ | −420.655(2) | $C_3H_{10}$ in $C_3H_{576}$ | −10.326(1) | 0.53 ± 0.62 |
| $N_{64}$ | −260.457(4) | $CH_4OH$ in $COH_{124}$ | −14.007(1) | −5.99 ± 0.15 |
| | | $OH_3$ in $OH_{124}$ | −12.977(1) | −7.39 ± 0.14 |
| | | $NH_4$ in $NH_{124}$ | −11.665(1) | −3.60 ± 0.15 |

For carbon at 500 GPa, the reference state is diamond. For oxygen, we use the C2/m space group structure[157] which is the most stable form from 100 to 250 GPa. For nitrogen, we use the helical tunnel structure of the $P2_12_12_1$ space group reported stable above 320 GPa[158]. Our calculations imply complete solubility of carbon, oxygen and nitrogen.

The positive heat of solution for carbon has been taken to suggest that carbon will condense and fall as diamond rain[86]. However, under the conditions expected in gas giants with metallic hydrogen cores, our calculations suggest a solubility limit above the expected primordial carbon-hydrogen ratio. Therefore, we anticipate that in many gas giant planets a significant proportion of the carbon will be found in solution in metallic hydrogen. Oxygen and nitrogen, as well as other elements[88], also have much higher solubilities in metallic hydrogen than in molecular hydrogen.

Although detailed planetary models are beyond the scope of this work, the consequence of negative enthalpy of mixing is that there may be no rocky cores of giant planets. Heavy elements may simply be mixed into the metallic hydrogen layer, forming an undifferentiated metallic region. The most recent data from the Juno mission[124] probed the core by measuring gravitational and magnetic fields, which suggested that the core is more extended and dilute than expected from previous models based on solubility from the molecular region[4,125,126]. Various explanations of the "fuzzy-core" have been advanced based on the history and formation kinetics of giant planets[126], but our work suggests that such a structure may simply follow from the equilibrium thermodynamics of solubility, acting in competition with gravitational forces to produce a concentration gradient of heavy elements in the core. The ongoing JUICE mission is focused on icy moons and the atmosphere and is not designed to provide much further information about the Jovian interior, although more detailed magnetosphere measurements and the tidal $Q$[127] may help to determine the existence or otherwise of a rocky core[128].

Experimental verification of these molecules on Earth is challenging but within the reach of current methodology. Liquid metallic hydrogen forms at lower pressures than its solid counterpart, and has been observed in both static and dynamic compression[8,70,129,130]. So, creation of hypermolecules is plausible, but detection of bonding in high pressure hydrogen is difficult[131–133]. Our $CH_6$ calculation suggests that the molecules will have distinctive vibrational modes at higher frequency than the atomic hydrogen frequencies, but lower than, e.g. $CH_4$.

Two "pathways" to metallic hydrogen[134] exist: via high temperature or via high pressure. Thermodynamically, these lead to the same atomic fluid phase[67], although at high temperature the higher entropy of atoms vs molecules dominates, whereas at pressure the denser metallic aspect is more important. A very recent paper[49] has studied the complementary situation to this work: very high temperatures where they find $CH_4$ in metallic hydrogen. The difference is readily understandable in terms of the much lower electron density along the "high-T" pathway, which disfavours $CH_6$. It also appears that the $CH_4$ solubility is primarily driven by entropy, whereas the hypermolecules are enthalpically favoured by metallicity.

Since all our simulations produce long-lived hypermolecules, it seems certain that more complex molecules will also be stable. Thus, it appears that the metallic hydrogen environment, the most common state of condensed matter in the universe, is capable of supporting its own rich molecular chemistry.

## Methods
### DFT computational details
Calculations were performed using density functional theory (DFT), Born-Oppenheimer molecular dynamics (BOMD), jellium and Mulliken charge (Supplementary Tables 4–10) calculations[135,136] implemented in Quantum Espresso (QE)[137,138], and CASTEP[139]. Crystal Orbital Hamilton Population (COHP) was implemented in the *LOBSTER* package[140-142]. The four-atom conventional $I4_1/amd$ structure of atomic metallic hydrogen and diamond (for carbon) were fully optimized[143-145] at 500 GPa, employing a force convergence criterion of $10^{-5}$ eV/ Å and a fine Monkhorst–Pack grid $k$-mesh. The exchange-correlation functional used was GGA-PBE[146]. For QE phonon analysis, we optimized a snapshot of $CH_{124}$ and $H_{128}$ using norm-conserving pseudopotentials[147,148]. The eigenvalues and eigenvectors were computed based on density functional perturbation theory (DFPT)[149]. For COHP analysis, we repeated the calculations using projector augmented wave (PAW) pseudopotentials[150], as they are compatible with *LOBSTER*. No significant pseudopotential effects were observed. For the BOMD simulations, a time step of 0.5 fs was used with velocity-Verlet integration[151]. The isothermal-isobaric ensemble (NPT)[152] was employed, utilizing the Parrinello-Rahman barostat[153]. Additionally, the temperature was controlled at 300 K using the Berendsen thermostat[154]. Solubility calculations, both static relaxation and NPT MD, were based on a 4 × 4 × 2 supercell (128 hydrogens) of the four-atom conventional $I4_1/amd$ structure, with some hydrogens substituted by carbon (Supplementary Fig. 1).

For the liquid simulations, we use the NPT ensemble at density and system-sizes equivalent to the I4₁/amd containing various atomic compositions including $CH_{124}$, $CH_{276}$, $C_2H_{120}$, $C_2H_{276}$, $C_2H_{576}$, $C_3H_{276}$, $C_3H_{576}$, $COH_{124}$, $OH_{124}$, $NH_{124}$, $8CH_{992}$, and $4C_2H_{992}$. Here, $8CH_{992}$ and $4C_2H_{992}$ represent eight mono-carbon atoms and four dimer carbon molecules in metallic hydrogen, respectively. In the case of $C_3H_{276}$, we found that the $C_3$ chain dissociates, forming $C_2H_8$, and $CH_6$. In contrast, for $C_3H_{576}$, no dissociation of the $C_3H_{10}$ molecule was observed. Our BOMD simulations were performed at various temperatures and 500 GPa with up to 20,000 steps (10.0 ps) and checked for convergence of potential energy. We also performed shorter, larger simulations of around 1000 atoms which previous work[63] has shown sufficient to converge the RDF of metallic liquid hydrogen.

### Analytical methods of DFT results
To analyse the results, we consider the Radial Distribution Function (RDF) which can be calculated using

$$g(r) = \frac{1}{N\rho} \sum_{i=1}^{N} \sum_{k \neq i} \langle \delta(r + r_k - r_i) \rangle, \qquad (2)$$

and mean squared displacement (MSD) as computed using the ensemble average given by

$$\text{MSD}(t) = \langle |\boldsymbol{x}(t) - \boldsymbol{x}(0)|^2 \rangle. \qquad (3)$$

A flat MSD corresponds to a crystal structure with no diffusion. An MSD increasing linearly with time corresponds to brownian motion: typically this implies a liquid, but if the gradient is small it can also arise from a defect migrating in a solid: this in fact occurs when too many hydrogens are removed to inserted the carbon atom - a vacancy defect is created which migrates freely through the lattice.

We compute the phonon density of states (PhDOS) by performing a Fourier transformation of the velocity autocorrelation,[108–110,155] from the trajectory of each atom in AIMD given by,

$$D(\omega) = \int e^{i\omega\tau} VAC(\tau) d\tau, \qquad (4)$$

where $\tau = t - t\prime$, and the velocity autocorrelation (VAC) can be obtained by using

$$VAC(\tau) = \frac{\sum_{t\prime} \langle \boldsymbol{v}(t) \cdot \boldsymbol{v}(t\prime) \rangle}{\sum_{t\prime} \langle \boldsymbol{v}(t\prime) \cdot \boldsymbol{v}(t\prime) \rangle}, \qquad (5)$$

where the "initial velocity" $t\prime$ allows for the time-averaging over the simulation to enhance the ensemble average. We also show the comparison between phonon density of states from this method and DFPT calculation (Supplementary Fig. 9).

We find the solubility limit by equating the Gibbs free energy in the mixture with that in the pure substances I4₁/amd hydrogen and diamond carbon at 500 GPa[156] (see Supplementary Fig. S2 and the procedures in the supplementary).

$$G_{xy}(P,T) + k_B T \left[ c \ln c + (1-c) \ln(1-c) \right] = x G_H(P,T) + y G_C(P,T), \qquad (6)$$

where $c = y/(x+y)$ is the carbon concentration and $y = 1 - x$ is the atomic fraction of C.

Our initial analysis is based on partial RDFs of CH and OH separations (Fig. 3a). This shows that simulations of around one hundred or one thousand atoms gives the same hypermolecule formation, both number of bonds and bondlength (Fig. 3b). The geometry of these molecules is also confirmed (Fig. 3e–j) by the angle distribution as provided in the supplementary material. We observe liquid structure peaks that extend to the hundred atom unit cell size, however, within a thousand-atom simulation, this oscillating structure has decayed away exponentially (Fig. 3a) and (Supplementary Fig. 10). Therefore, we are confident that the results from our hundred-atom simulations give a good description of the hypermolecules.

## Data availability

Data for the figures are available at https://doi.org/10.6084/m9.figshare.28655204. Raw simulation output files and large intermediate datasets cannot be made publicly available due to their size and non-standard format. These data are available on request from the authors.

## Code availability

First-principles density functional theory (DFT) calculations were carried out using the CASTEP code https://www.castep.org/and Quantum ESPRESSO package https://www.quantum-espresso.org. For chemical bonding analysis, the LOBSTER code was employed http://www.cohp.de/.

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

## Acknowledgements

This research project is supported by the Second Century Fund (C2F), Chulalongkorn University. G.J.A. acknowledges funding from the ERC project Hecate. This work used the Cirrus UK National Tier-2 HPC Service at EPCC (http://www.cirrus.ac.uk) funded by the University of Edinburgh and EPSRC (EP/X035891/1). This also work used the ARCHER2 UK National Supercomputing Service (https://www.archer2.ac.uk) as part of the UKCP collaboration. We acknowledge the supporting computing infrastructure provided by NSTDA, CU, CUAASC, NSRF via PMUB [B05F650021, B37G660013] (Thailand) (www.e-science.in.th). We thank Pattanasak Teeratchanan, David Ceperley and Jeffrey M. McMahon for their valuable suggestions on DFT(QE)-related issues for studying metallic hydrogen. We thank Miriam Pena-Alvarez, Andreas Hermann, Miguel Martinez-Canales, and Stewart McWilliams for comment and proofreading.

## Author contributions

Jakkapat Seeyangnok performed the calculations, analysed the results, and wrote the first draft of the manuscript. Udomsilp Pinsook contributed to the analysis and data interpretation, and manuscript writing. Graeme J. Ackland supervised the project, contributed to data interpretation, coordinated the research activities, and assisted in writing the manuscript. All authors reviewed and approved the final version of the manuscript.

## Competing interests

The authors declare no competing interests.
