## [Transparent Peer Review file · Nature Communications]

Hydrogenation of saturated organic and inorganic molecules in metallic hydrogen

Corresponding Author: Professor Graeme Ackland

Version 0:

Reviewer comments:

Reviewer #1

(Remarks to the Author)

In this paper the authors study the effect of organic compounds in the background of metallic hydrogen. Molecules of the C-H-O complex (as well as N) form a large fraction of the interior of ice giants like Uranus and Neptune in our solar system and of warmer Neptune-like exoplanets detected in great number over the last decades. Water, methane, and ammonia molecules will dissociate and the resulting atoms be ionized when pressure and temperature increase towards the center of these planets. The calculation of the electronic structure dependent on pressure and temperature in the presence of excess hydrogen which will be metallic at such extreme conditions beyond 1-2 Mbar represents a major challenge to computational physics.

First, the authors give a broad introduction into high-pressure physics, related aspects of planetary physics, and the computational challenges. Unfortunately, the numerous references with respect to planetary physics (gas giants and ice giants) are a bit outdated since standard (but too simplistic) three-layer models can no longer reproduce the density profiles through Jupiter and Saturn as shown by the gravity data provided by the Juno and Cassini missions. H₂-He demixing and the solubility of higher-Z core material in metallic hydrogen yield strong compositional gradients and possibly stratified layers resulting in non-adiabatic P-T profiles. The same is expected for the interiors of ice giants due to H₂-H₂O demixing and again core erosion. For a recent comprehensive review, see e.g. <https://arxiv.org/abs/2205.04100>

Specifically, the authors study the solubility of organic molecules in metallic hydrogen and hypothesize to have found new molecular structures which they call “hypermolecules”. If their prediction is confirmed these transient structural entities could have strong implications in understanding the properties of ice giants in the solar system and beyond as the chemistry of molecules under those extreme conditions is currently not well understood.

However, some question about their results arise after reading the manuscript. We give our comments below.

1) Introduction paragraph “Modern planetary models depict layers separated by the weight of elements.....” the reference should occur before the preposition. A general criticism with respect to the literature with respect to the interior structure of gas and ice giants is given above.

2) The use of the word “covalent bond”, which involves the sharing of electrons, should be used carefully in the article and needs more justification. C [1s² 2s² 2p²] can only have four covalent bonds due to the unoccupied “2p” (let’s assume p_y and p_z) orbital. To achieve six covalent bonds with a carbon atom the whole “2p” orbital has to be empty. Can the authors shed more light onto the ionization of the carbon atom in their study and if the charges of “hypermolecules” are getting stabilized due to metallic hydrogen, and how can this complex electronic configuration represent a covalent bond? This is contradictory from our perspective. Can the authors clarify this point?

3) What is the reason for the specific size of the supercells chosen in this study? Did the authors perform an analysis dependent on different particle numbers N?

4) Did the authors carry out on-site analyses? I.e. the lowest energy structure concerning the position of carbon in metallic hydrogen (l41/amd)? (in static DFT calculations)

5) The labels inside all figures are too small. We suggest to increase their size.

6) Caption of Fig. 1 "... b) Electronic density of states from a typical BOMD snapshot, showing the characteristic free electron form with a pseudogap ...". For a DFT-MD simulation the usual practice is to calculate electronic properties averaged over the whole equilibrated trajectory. Here the authors provide the electronic density of state just for a single snapshot (?) which is not conclusive at all. The same holds also for the ELF shown in panels d-f.

7) The same question as raised under 5. is even more relevant for the phonon DOS shown in Fig. 1c. The authors claim that the peak at 3163 cm^{-1} is due to "hypermethane". Results from a single snapshot are not at all conclusive, based on an arbitrary ion configuration and the corresponding forces. Performing averages for observable quantities over the whole (equilibrated) simulation trajectory is essential for warm dense matter. Results based on single snapshots are not conclusive or, strictly speaking, useless. Could the authors provide lattice dynamics and electronic properties averaged over equilibrated BOMD trajectories?

8) The authors refer to these "hypermolecules" in their simulation but we have not found conclusive evidence for these complexes. Radial distribution function and coordination number analyses are reliable tools to analyze the structure of the system, giving the probability and number of ions surrounding the reference atom, respectively. However, it is not enough to back their claim. Can the authors show, e.g., a three-body distribution, a bond-angle analysis, or a Mulliken analysis to verify their claim?

9) Fig. 3. The figures in panels a-d are much too small and not recognizable. The same is true for the other panels. An increase is necessary for the readability of the manuscript.

10) Fig. 3 Panels e-f-g-h-i-j need more justification as the figures of the molecules are user-defined, i.e., the length of the bond can be modified in the visualization software (seems like VESTA) which the authors have used. One should be careful using these figures as proof for the existence of "hypermolecules". It seems that they represent again only a single snapshot of the MD simulation and not an average over a complete MD trajectory.

We conclude that the results and explanations provided in the manuscript are not conclusive enough to recommend publication in Nature Communications.

Reviewer #2

(Remarks to the Author)

The paper is interesting, but requires substantial rethinking and modification before it can be considered further.

1. The solubility of hydrocarbons in solid and liquid hydrogen should be very different and to claim a role in planets, one needs to have serious estimates of the solubility at least in liquid H.
2. In most of the paper, negative charge is claimed (but never demonstrated) for these molecules, but then on p. 9 the authors suddenly assign positive charges to CH_6 , H_3O , NH_4 . Which is correct?
3. The explanation that added electrons enable more bonds is chemically unreasonable, as added electrons occupy in all these molecules antibonding orbitals and will lead to bond breaking.
4. Generally, increased coordination under pressure is normal, as in SiO_4 to SiO_6 change. Drawing chemical diagrams with 6 single C-H bonds is incorrect too. Here, bonds become fractional.
5. Literature is cited sporadically (on silicates) and lacking recent research (on hydrocarbons under pressure).

Major revision.

Reviewer #3

(Remarks to the Author)

Reviewer #4

(Remarks to the Author)

The study uses density functional theory to characterize C-H bonding and molecule stability for conditions at which hydrogen is a metallic fluid; O-H and N-H bonding are also considered. Addressing chemistry at metallic-hydrogen conditions ought to be of interest to many disciplines, including chemistry, materials physics and planetary science.

The point of the paper is unclear, however, and it is likely to have minimal impact as currently written. First, it contains typographical errors, jargon and unclear material that make it hard for a nonspecialist to understand the assumptions, limitations and clearcut results of the work being presented. Fig. 3 is too small to read, and it is not obvious why 10 ps and $4 \times 4 \times 2$ cells are adequate, for example. Second, the lack of hypotheses being tested or even proposed makes it hard for the reader to know what to make of this paper. The final section (Discussion and Conclusions) is notably confusing in acknowledging that no clear implications are being inferred for planets, nor is any experimental validation being proposed.

In summary, the underlying work seems OK (though its reliability ought to be more thoroughly justified), but it is written up in a manner more suitable for a journal aimed at specialists rather than a general audience. It would help if the authors were to clarify whether they are describing a new kind of organic chemistry (the latter being defined as consisting of C-H-O-N compounds), or are demonstrating that C-H-O-N no longer form organic compounds at the conditions being considered (no longer characterized by the bonding and polymer formation of organic chemistry at ambient conditions). In either case, the significance of the results should be clearly stated.

Reviewer #5

(Remarks to the Author)

Version 1:

Reviewer comments:

Reviewer #2

(Remarks to the Author)

The manuscript has improved, although I am not fully satisfied with the authors's replies and still see major issues to be addressed (see below).

1. The calculations of enthalpic effects of incorporating impurities into pure hydrogen are not fully clear. Tables give as-calculated enthalpies, which are of no interest: the interesting quantity is the enthalpy of mixing which requires subtraction of enthalpies of pure compounds (pure H and pure C, and one should also try pure hydrocarbons stable at relevant conditions).
2. The ratio C/H on Jupiter is much greater than 1 ppm and is in the range 0.1-1%. If 1 ppm of C is dissolved in hydrogen, then most carbon won't be in this form.
3. I am not sure what happened in the calculation, but strongly positive enthalpy of dissolution of C (0.39 eV) then changes into strongly negative values when discussing the incorporation of hydrocarbon molecules, and this is also problematic. The reason is that the formation of solutions is related to positive values of mixing energy (but if it is too large, solubility will be very small). Negative values of mixing enthalpy indicate the formation of compounds. The authors' conclusion of complete solubility (based on negative mixing enthalpy) is incorrect - what follows is that some amount of carbon will react and form some hydrocarbon, and that hydrocarbon at low T will form a separate solid phase and at high T may or may not mix with liquid hydrogen. This really needs to be investigated.
4. The formation of hypermolecules is indeed interesting, although in pure hydrocarbons this was not predicted, as far as I remember. Could this be a consequence of the hydrogen matrix?
5. Conclusions of 6-valent carbon are toned down in the current manuscript, and should be removed altogether. Carbon has 4 valence electrons and cannot have valence greater than 4. To enable hexavalent state, you need to remove two electrons from carbon - then, C²⁺ has the ability to attract six electrons. Here it's very hard to see how this would happen. So one should only talk about increased coordination number. As a check, you can compare C-H distance in CH₆ with that in CH₄ at the same pressure (the distance in CH₄ should be shorter if my reasoning is correct).

So the final conclusion is: major revision is needed.

Reviewer #3

(Remarks to the Author)

I have assessed the author response and changes in the manuscript. The authors have addressed with the previous concerns and produced additional data to support their findings. They have also made changes in the manuscript (specially adding a new section) and figures are much more visible. I think the manuscript is in better position than before, and I have no additional comments to the authors.

Reviewer #4

(Remarks to the Author)

The paper is much improved over the original. The additional work done in response to reviews helps, and the presentation has been improved with more legible figures and text. Although the authors have addressed Reviewer 1 and 2's more narrow, technical comments, the broader concerns remain to be addressed.

1) It is unclear what the authors mean by "organic compounds" and "organic-style molecules" (title and abstract). In response to review comments, they have now done a better job documenting covalent bonding (localized electron density between ions) and hybridization in their simulations (p. 9). It is unclear what these characteristics have to do with organic chemistry as generally understood, for example with its rich diversity of polymeric structures.

Because it does not clarify, yet has the potential for causing misunderstanding, the authors should consider dropping the term "organic" in their title and text unless they can provide substantive reason for using it (they might consider using "covalent" instead). The hope is that publication of this work will not result in a press release leading people to think that metallic hydrogen might provide a habitat for life.

2. The analogy between the structures of C- and Si-based oxides at high pressures has been documented through experiments, not just through computer simulations as in the present work. The authors ought to address the work of Yoo, Iota and colleagues on this matter [see Phys. Rev. B 82, 012105 (2010) and references 1-9 therein; see also the 2020 review at doi: 10.1063/1.5127897], with C-O compounds being more relevant to the issue than is the analogy with C-H compounds that they discuss.

As a detail, it is unclear what the authors mean by "pressure-unfavorable longer bondlengths" (p. 9, ¶ 3), as pressure tends to increase first-neighbor coordination numbers, which increases bond lengths (first-neighbor distances) both in fluids and crystals.

3. The authors' conclusions (e.g., p. 4 near bottom, p. 9 near top) of various elements being soluble at the 10^0 - 10^2 ppm level in metallic hydrogen (i.e., for Solar/Jovian abundances of all elements other than He) is hardly surprising or useful, as it is difficult to avoid this level of solubility due to the entropy of mixing in any fluid at planetary interior temperatures.

4. Perhaps most disappointing in the response to reviews and in the revised text is that the authors indicate no way of empirically validating or falsifying their results, whether through experiments or observations on planets.

Reviewer #5

(Remarks to the Author)

Reviewer #6

(Remarks to the Author)

I have carefully read the revised manuscript and the reports of reviewers no. 2 and no. 4 on the latest version. I share the scepticism expressed by both reviewers. On the other hand, I think the topic is interesting enough for publication in Nature Communications. The authors should be given another chance to make changes and/or add to the paper before a final decision is made.

Reviewer #7

(Remarks to the Author)

This Referee will be only concerned about the following issues:

1.) Is 6-coordination of carbon atoms something we do not know in organic chemistry?

The clear answer is no. Non-classical carbo cations is the keyword. The recent publication (Malischewski, Seppelt, Angew. Chem. Int. Ed. 2017, 56, 368) reports on covalently six-coordinated carbon species in pentagonal pyramidal molecules $[C_6(CH_3)_6]^{2+}$, where each C atom has an exo ligand CH_3 ; 5 C atoms form a ring capped by the 6th carbon atom. This stable cluster is equivalent to a nido borane B_6H_{10} and well understood from the Lipscomb's work (who synthesized the cluster in 1954, determined its crystal structure 1958, and explained its bonding pattern, see also his noble lecture (Fig. 14 in Science 1977, 196, 1047ff) and the Wade's rules (1971).

2.) Do the ELF plots provide evidence of covalent bonds between central C and neighbor H atoms in $CH_6@H(\text{metal})$?

The clear answer is no. The 'smoking gun' feature of a covalent chemical bond in the framework of ELF analysis of chemical bonding is the cre

ation of a local maximum (attractor) between the atoms involved. This is not seen in the ELF diagrams shown (please avoid calling them 'ELFs'; for the same reason you were not writing that you were doing a couple of 'DFTs' to investigate the energetic situation of C in metallic H).

The attractors seen there are at the position of H nuclei, which is unfortunately something you always see on H in any compound. Since ELF shows shell structure, it gives a maximum for the 1st shell of H, which accidentally is also the valence region. This way also in classical et

hane C_2H_6 the C-H bond does only show a maximum at the H position related to the shell structure already, while the C-C bond shows a new attractor between the carbon atoms. This is the ELF proof a C-C bond here. The high values between C and H seen in the ELF diagrams Fig. 1 and SI

Fig. 6 of the ms. do not indicate a special type of covalent bonds; if you look into literature

(Savin, Nesper, Wengert, Faessler, ELF: The Electron Localization Function, Angew. Chem. Int. Ed. 1997, 36, 1808) you see in Fig. 2a a plot of ELF for ethane with the C-C bond attractor, and also in Fig. 2c the ELF distribution between two non-bonded H atoms at the same C atom in C_2H_4 ; the values are quite high there like in the diagrams shown for $CH_6@H(\text{metal})$. A remark on the ELF diagrams shown: The core (1s) region of C has been omitted (why?). It would be interesting to see this region for a compressed situation with such a high pressure. Does C(1s) density leak out into the valence region?

3.) Concerning section 2.4 Chemistry of hypermolecules:

The decisive sentence seems to me "... analysis whether the bonds are fractionally occupied is complicated because the bonding states are degenerate with free electrons." This clearly indicates that the 'molecules' are not at all isolated (like CH₄ in our atmosphere of N₂, and O₂) but embedded in the 'H matrix' like an arbitrary fragment of an alloy is. I would tend to believe that COHP results indicate that C is covalently bonded to 6 H atoms of this matrix. However, and this has not been investigated, I guess the ligating H atoms would be found to be still bonded to the matrix but slightly less than the undisturbed H atoms of the matrix. Thus, there are no separate CH₆ molecules in this matrix. To my opinion it does not help nowadays going to simpler models like the jellium one if we have not understood it with first-principle methods like DFT.

Conclusion: I really like the study and the computational work, but the bonding discussion is not at the same high level as the remaining part. I cannot see molecules like CH₆ as separated entities in the metallic H matrix. Moreover, six-coordinated C atoms are known in organic chemistry since the second half of the 20th century, which may destroy the quest of hype.

Version 2:

Reviewer comments:

Reviewer #2

(Remarks to the Author)

I am satisfied with the revised manuscript and suggest its acceptance

Reviewer #4

(Remarks to the Author)

Reviewer #5

(Remarks to the Author)

Reviewer #7

(Remarks to the Author)

The revised manuscript has considered my remarks in a sufficient way.

I do not want to open further discussion about chemical bonding in these compounds because the authors have been done enough to substantiate their ideas. Of course, this will stimulate discussion and further investigations using different techniques, which is even an effect implicitly or explicitly intended by such a study.

The only thing that disturbs me as a chemist is the classification of H₂O and NH₃ as organic molecules. Of course, the molecules cannot defend themselves against this (i.e., it is not a measurable property), but a look into organic and inorganic chemistry textbooks (at least those that I know) will show that these molecules are treated in the inorganic chemistry ones.

The title of the study may be interpreted as being related to only a part of the molecules actually computed, which I feel is ok. However, in the abstract "organic-style molecules OH₃, NH₄" are explicitly mentioned, which is unconventional at least. It would not harm the interest in and the significance of the results if these two molecules were more conventionally classified. It will even show that not only saturated "organic molecules" become hydrogenated but also inorganic ones. The chemically important and new issue is that saturated molecules like CH₄ and C₂H₆ (see below) get hydrogenated, while it is common knowledge (freshmen courses of organic chemistry) that this is possible only for unsaturated hydrocarbons, e.g. 2HC=CH₂ + H₂ -> 3HC-CH₃.

Summarizing, I have nothing more to complain, just a few friendly recommendations left free to follow for the authors.

Dear Editors

NCOMMS-24-44714-T

Organic compounds in metallic hydrogen

Thank you for the positive reports from all the referees. We found their comments helpful, but to fully address them required a large amount of additional computations which we have only now completed - apologies for the delayed reply.

In particular, to address the requirements to present a "chemistry"-style explanation of the hypermolecules we had to use Extended Huckel theory: the standard code for this, LOBSTER, is incompatible with our DFT code CASTEP, so we had to repeat some of the DFT calculations with a different DFT code.

We also carried out additional calculation to get ensemble averages in place of single snapshots, and some jellium calculations to disentangle the effects of pressure and metallicity.

Put together, the results of these new calculation are remarkably convincing in underlining the chemical nature of the hypermolecules as covalently bonded molecular ions. So we included them in the main paper as an additional section which provides a clearer explanation for readers with a chemistry background.

We have addressed all inquiries and have revised the manuscript accordingly, based on the referees' comments. New sections are shown in colour.

We hope that the work is now suitable for publication in Nature Communication.

Sincerely yours,

J.Seeyangnok, U.Pinsook, and G.J.Ackland

Reviewer #1 (Remarks to the Author):

In this paper the authors study the effect of organic compounds in the background of metallic hydrogen. Molecules of the C-H-O complex (as well as N) form a large fraction of the interior of ice giants like Uranus and Neptune in our solar system and of warmer Neptune-like exoplanets detected in great number over the last decades. Water, methane, and ammonia molecules will dissociate and the resulting atoms be ionized when pressure and temperature increase towards the center of these planets. The calculation of the electronic structure dependent on pressure and temperature in the presence of excess hydrogen which will be metallic at such extreme conditions beyond 1-2 Mbar represents a major challenge to computational physics.

First, the authors give a broad introduction into high-pressure physics, related aspects of planetary physics, and the computational challenges. Unfortunately, the numerous references with respect to planetary physics (gas giants and ice giants) are a bit outdated since standard (but too simplistic) three-layer models can no longer reproduce the density profiles through Jupiter and Saturn as shown by the gravity data provided by the Juno and Cassini missions. H₂-He demixing and the solubility of higher-Z core material in metallic hydrogen yield strong compositional gradients and possibly

stratified layers resulting in non-adiabatic P-T profiles. The same is expected for the interiors of ice giants due to H₂-H₂O demixing and again core erosion. For a recent comprehensive review, see e.g. <https://arxiv.org/abs/2205.04100>

Specifically, the authors study the solubility of organic molecules in metallic hydrogen and hypothesize to have found new molecular structures which they call “hypermolecules”. If their prediction is confirmed these transient structural entities could have strong implications in understanding the properties of ice giants in the solar system and beyond as the chemistry of molecules under those extreme conditions is currently not well understood.

However, some question about their results arise after reading the manuscript. We give our comments below.

1) Introduction paragraph “Modern planetary models depict layers separated by the weight of elements. . . .” the reference should occur before the preposition. A general criticism with respect to the literature with respect to the interior structure of gas and ice giants is given above. **Answer: We have changed the position of the reference.**

2) The use of the word “covalent bond”, which involves the sharing of electrons, should be used carefully in the article and needs more justification. C [1s² 2s² 2p²] can only have four covalent bonds due to the unoccupied “2p” (let’s assume py and pz) orbital. To achieve six covalent bonds with a carbon atom the whole “2p” orbital has to be empty. Can the authors shed more light onto the ionization of the carbon atom in their study and if the charges of “hypermolecules” are getting stabilized due to metallic hydrogen, and how can this complex electronic configuration represent a covalent bond? This is contradictory from our perspective. Can the authors clarify this point?

n.b. Referee 2 asked a similar question in points 2 and 3

Defining a covalent bond in a metallic background based on a plane wave basis and Kohn-Sham orbitals set requires care. In a recent paper by Marques et al this was discussed at some length in the context of metal hydrides. Even a simple calculation of a single methane molecule yields one low energy s-type orbital and three higher energy p-type, rather than four equivalent “bonds”: this follows directly from the Td symmetry.

This point is sufficiently central to our work that we carried out many further calculation to illustrate the point, leading to a new section of the paper.

In the original manuscript, we identified bonds by localised charge between pairs of atoms, and high values of the electron localisation function ELF. In the case of CH₆ we identified six, equivalent CH bonds. Assigning charge to atoms in a free electron metal is rather ambiguous, but six bonds would require a charge of -2, which would in turn be screened by the electrons leading to the observed Friedel oscillations. For certain, there are six persistent bonds and they are equivalent: Reviewer 2 suggests that they should be “become fractional” so that the hypermolecule can remain neutral.

For a more chemical picture, we also conducted a Crystal Orbital Hamilton Population

(COHP) analysis. The COHP method provides a detailed understanding of bonding and anti-bonding interactions by quantifying the energy contributions from orbital overlap within a chemical bond. In this study, we use conventional -COHP diagrams, where bonding states are represented as positive values (to the right) and anti-bonding states as negative values (to the left). The results, shown in the figure below and in Supplemental materials, indicate that the projected Crystal Orbital Hamilton Population (-pCOHP) on the s and p orbitals for the hydrogen and carbon atoms is positive below the Fermi energy (valence electrons), confirming the presence of bonding interactions. Notably, the results suggest orbital mixing ("hybridization") between the s and p orbitals, which is similar to the typical hybridization observed in hydrocarbon molecules under ambient conditions.

Figures show projected electronic bands, density of states, and -pCOHP of CH_6 in metallic hydrogen background.

To compare with hydrogen atoms in the system that do not bond with the carbon, we calculated the -pCOHP for two additional hydrogen atoms located at distances of 1.97 Å and 3.70 Å. As shown in the figure below, the -pCOHP diagram indicates no bonding character, as there is no significant overlap between the wavefunctions of these hydrogen atoms and the central carbon atom.

Figures show the $-p\text{COHP}$ among CH_6 and two additional hydrogen atoms.

Within the DFT framework, we calculated the stability and electronic structure of CH_6 compared to $\text{CH}_6 + \text{H}_2$ in a free electron gas (jellium). This demonstrates how there are electronic states localised on the CH_6 which have energies well below the Fermi level, but that these states always lie within the free electron band and are therefore hybridised with the free electrons

3) What is the reason for the specific size of the supercells chosen in this study? Did the authors perform an analysis dependent on different particle numbers N ?

Answer: MD requires large enough number of atoms and long enough timescales to get sensible results. We chose 442 supercell (128 atoms) to allow enough simulation time for bond making and breaking. We also tested our result with a 1000-atomic system to verify that the rdf structure was converged. We have the rdfs to give some idea of the lengthscale of the structure around the molecules. For the solubility calculations, the ZPE is crucial, and the error from sample noise in the ZPE increases with system size. So that set a maximum size.

4) Did the authors carry out on-site analyses? I.e. the lowest energy structure concerning the position of carbon in metallic hydrogen (I41/amd)? (in static DFT calculations)

Answer: Yes, we did test static relaxation from the different positions of carbon substitution in I41/amd. We have also computed the enthalpy of the relaxed structures from four different MD which gives the same value to within 0.1 meV of -10.98386 eV/atom. This compares to the static relaxation of -10.93038 eV/atom, suggesting that there are some additional symmetry-breakings.

5) The labels inside all figures are too small. We suggest to increase their size.

Answer: We have increased the size of the labels inside all figures.

6) Caption of Fig. 1 “. . . b) Electronic density of states from a typical BOMD snapshot, showing the characteristic free electron form with a pseudogap ...”. For a DFT-MD simulation the usual practice is to calculate electronic properties averaged over the whole equilibrated trajectory. Here the authors provide the electronic density of state just for a single snapshot (?) which is not conclusive at all. The same holds also for the ELF shown in panels d-f.

Answer: We have checked a number of snapshots and presented a representative one. they are very similar because the e-DOS is rather featureless and the system large and well-sampled. But we agree that a single snapshot risks being perceived as unrepresentative by the reader. To demonstrate, in the revised Fig 1b we show the DoS from four different snapshots and the average value of the four snapshot along with the DoS of iA_1/amd metallic hydrogen. This makes it clear that they are well converged.

Figures show the DoS of CH_{124} for four different snapshots (blue, orange, green, and red) which correspond to each other and are broadly similar to the DoS of $I4_1/amd$ metallic hydrogen without carbon (dashed black).

Figures show the Electron Localised Function (ELF) of CH124 for four different snapshots. All ELFs indicate high values between the carbon and the six surrounding hydrogen atoms.

7) The same question was raised under 5. is even more relevant for the phonon DOS shown in Fig. 1c. The authors claim that the peak at 3163 cm^{-1} is due to “hypermethane”. Results from a single snapshot are not at all conclusive, based on an arbitrary ion configuration and the corresponding forces. Performing averages for observable quantities over the whole (equilibrated) simulation trajectory is essential for warm dense matter. Results based on single snapshots are not conclusive or, strictly speaking, useless. Could the authors provide lattice dynamics and electronic properties averaged over equilibrated BOMD trajectories?

As in 5, we have now computed the phonon frequency at the Γ point for four optimized structures from four different snapshots. We found that all of our calculations show the distinctive peak at 3161 cm^{-1} for the three calculations and $3,162 \text{ cm}^{-1}$ for the other. This remarkable consistency is strong evidence that the hypermethane is a well-defined entity and is now stated in the paper. We thank the referee for helping make the argument more convincing .

8) The authors refer to these “hypermolecules” in their simulation but we have not

found conclusive evidence for these complexes. Radial distribution function and coordination number analyses are reliable tools to analyze the structure of the system, giving the probability and number of ions surrounding the reference atom, respectively. However, it is not enough to back their claim. Can the authors show, e.g., a three-body distribution, a bond-angle analysis, or a Mulliken analysis to verify their claim?

Answer: We have calculated the average position of CH₆ in *I4/amd* which is shown below. Since the same six hydrogens are bonded to the carbon throughout the simulation, it is also straightforward to get the angular distribution function between hydrogens and carbon are listed in the table. We also calculated the Mulliken charges on the hypermolecules. These show that, as the referee imagined, there are well-defined bond angles in all hypermolecules. We placed the ADF for CH₆ in the paper, and the remaining figures and tables in supplementary information

Figures show the average positions of CH₆ in solid metallic hydrogen at 300K where X, Y and Z are position of atoms in atomic unit.

Atomic types	Angles (degree)
1H and 2H	87.02
1H and 3H	93.30
1H and 4H	81.97
1H and 5H	98.72
1H and 6H	178.03
2H and 3H	178.31
2H and 4H	86.88
2H and 5H	90.91
2H and 6H	91.16
3H and 4H	94.81
3H and 5H	87.40
3H and 6H	88.50
4H and 5H	177.65
4H and 6H	98.69
5H and 6H	80.54

Table shows the average angles of CH₆ in solid metallic hydrogen at 500GPa and 300K.

We also computed the Mulliken charge from different snapshots the result is shown in the table below.

CH ₆ in solid hydrogen	Mulliken 1 (-e)	Mulliken 2 (-e)	Mulliken 3 (-e)	Mulliken 4 (-e)
C	-0.358	-0.371	-0.382	-0.358
H	0.068	0.091	0.062	0.077
H	0.083	0.071	0.065	0.072
H	0.058	0.058	0.056	0.062
H	0.067	0.079	0.094	0.069
H	0.059	0.082	0.088	0.073
H	0.057	0.050	0.084	0.065

Table shows the Mulliken charge of from different snapshots of CH₆ in solid metallic hydrogen at 500GPa, 300K.

CH ₆ in liquid hydrogen	Mulliken 1 (-e)	Mulliken 2 (-e)
C	-0.367	-0.365
H	0.083	0.050
H	0.062	0.033
H	0.059	0.042
H	0.062	0.070
H	0.105	0.070
H	0.053	0.062

Table shows the Mulliken charge of from different snapshots of CH₆ in liquid metallic hydrogen background at 500GPa, 600K.

C ₂ H ₈ in liquid hydrogen	Mulliken 1 (-e)	Mulliken 2 (-e)
C1	-0.291	-0.248
H	0.059	0.036
H	0.084	0.070
H	0.083	0.070
H	0.059	0.059
C2	-0.278	-0.272
H	0.084	0.058
H	0.077	0.065
H	0.056	0.051
H	0.090	0.073

Table shows the Mulliken charge of from different snapshots of C₂H₈ in liquid metallic hydrogen background at 500GPa, 600K.

C ₃ H ₁₀ in liquid hydrogen	Mulliken 1 (-e)	Mulliken 2 (-e)
C1	-0.321	-0.295
H	0.091	0.073
H	0.097	0.059
H	0.076	0.077
H	0.107	0.071
C2	-0.173	-0.146
H	0.047	0.047
H	0.059	0.049
C3	-0.263	-0.277
H	0.050	0.066
H	0.076	0.067
H	0.105	0.071
H	0.068	0.048

Table shows the Mulliken charge of from different snapshots of C₃H₁₀ in liquid metallic hydrogen background at 500GPa, 600K.

CH ₄ OH in liquid hydrogen	Mulliken 1 (-e)	Mulliken 2 (-e)
C	-0.069	-0.081
H	0.065	0.063
H	0.073	0.077
H	0.053	0.066
H	0.050	0.062
O	-0.438	-0.442
H	0.261	0.163

Table shows the Mulliken charge of from different snapshots of CH₄OH in liquid metallic hydrogen background at 500GPa, 600K.

OH ₃ in liquid hydrogen	Mulliken 1 (-e)	Mulliken 2 (-e)
O	-0.519	-0.469
H	0.144	0.201
H	0.213	0.251
H	0.221	0.232

Table shows the Mulliken charge of from different snapshots of OH₃ in liquid metallic hydrogen background at 500GPa, 600K.

NH ₄ in liquid hydrogen	Mulliken 1 (-e)	Mulliken 2 (-e)
N	-0.415	-0.428
H	0.161	0.189
H	0.174	0.168
H	0.169	0.133
H	0.160	0.134

Table shows the Mulliken charge of from different snapshots of NH₄ in liquid metallic hydrogen background at 500GPa, 600K.

Figure shows the angle distribution between pairs of hydrogen atoms with respect to carbon for each step of MD within the radius cutoff of 1.30Å where the average number of coordination is six. Two significant peaks show expected octahedral angles at 90 and 180 degrees.

Figure shows the angle distribution between pairs of hydrogen atoms with respect to carbon for each step of MD within the radius cutoff of 1.30Å where the average number of coordination is six. Two significant peaks show probable angles at 90 and 180 degree with large smearing of the distribution resulting from high anharmonicity of hydrogen.

Figures show the angle distribution between pairs of hydrogen atoms with respect to each carbon of C₁ and C₂ separately for each step of MD within the radius cutoff of 1.30Å where the average number of coordination for each carbon is four. Two significant peaks show probable angles at 75 and 144 degrees.

Figures show the angle distribution between pairs of hydrogen atoms with respect to each carbon of C_1 , C_2 , and C_3 separately for each step of MD within the radius cutoff of 1.30\AA for C_1 and C_3 and 1.16\AA for C_2 . The coordination for C_1 and C_3 is four, and two for C_2 . Two significant peaks show probable angles at 75 and 144 degree with large smearing of the distribution resulting from high anharmonicity of hydrogen for C_1 and C_3 . The single peak shows a unique angle between two hydrogen atoms surrounding C_2 at around 104 degrees.

Figures show the angle distribution between pairs of hydrogen atoms with respect to carbon for each step of MD within the radius cutoff of 1.37\AA where the coordination is four. Two significant peaks show probable angles at 75 and 144 degrees similar to

the case of C_2H_8 and C_3H_{10} .

Figures show the angle distribution between pairs of hydrogen atoms with respect to oxygen for each step of MD within the radius cutoff of 1.32\AA where the average number of coordination is three. Single significant peak shows probable angle at 103° similar to the case of C_3H_{10} .

Figures show the angle distribution between pairs of hydrogen atoms with respect to oxygen for each step of MD within the radius cutoff of 1.20\AA where the average number of coordination is four. Single significant peak shows probable angle at 110° which corresponds to the angle of methane at ambient.

9) Fig. 3. The figures in panels a-d are much too small and not recognizable. The same is true for the other panels. An increase is necessary for the readability of the manuscript.

Answer: We have increased and improved the figures as suggested by the referee.

10) Fig. 3 Panels e-f-g-h-i-j need more justification as the figures of the molecules are user-defined, i.e., the length of the bond can be modified in the visualization software (seems like VESTA) which the authors have used. One should be careful using these figures as proof for the existence of “hypermolecules”. It seems that they represent

again only a single snapshot of the MD simulation and not an average over a complete MD trajectory.

Answer: As with points 5) and 7) referee is correct that a single snapshot does not constitute proof, however the shown snapshots are genuinely typical of every snapshot we have imaged and, we believe, are a good representation of the hypermolecules. We also note that each atom is indexed in the MD, and throughout all simulations, the same atoms comprise the hypermolecule. As a compromise, we added a second snapshot in each case. Although the bondlengths are user-defined in VESTA, the sharp first peak in the RDF provides the appropriate value. Furthermore, the same six hydrogens are the "neighbours" throughout the simulation, so choosing snapshots with just those six neighbours is not only typical, but also well justified.

Computing the average position of atoms in liquid metallic hydrogen is meaningless because of diffusion. To demonstrate the permanence we consider the properties of "hypermolecules" via ensemble average of rotational and translational invariant quantities such as the radial distribution function, the cumulative number of hydrogens, angle distribution and Mulliken charge properties. We also note that the hypermolecules contain the same hydrogen atoms throughout the simulation - there is no bond-breaking.

We conclude that the results and explanations provided in the manuscript are not conclusive enough to recommend publication in Nature Communications.

We thank the referee for pressing us to provide more details in order to demonstrate the reality of these hypermolecules. This has entailed a lot more work, but the manuscript is much stronger and better as a result.

We accepted the criticism that we should have presented more thorough statistical analysis of quantities such as DoS, RDF and ADF, and been clearer about the persistence of the atoms in the hypermolecules in the original manuscript. We have now done so, and the paper now better presents to confidence we have in the persistence and consistent structures of the hypermolecules. We have done extensive further calculations to relate the chemistry of the hypermolecules in the presence of excess free electrons to the conventional low-pressure chemistry of atoms. In particular, the COHP analysis demonstrates that the six neighbours in CH_6 have both occupied bonding and unoccupied antibonding type orbitals, while other nearby hydrogens are unbonded.

Reviewer #2 (Remarks to the Author):

The paper is interesting, but requires substantial rethinking and modification before it can be considered further.

1. The solubility of hydrocarbons in solid and liquid hydrogen should be very different and to claim a role in planets, one needs to have serious estimates of the solubility at least in liquid H.

Answer: We believe that it is most likely that carbon, nitrogen and oxygen are fully soluble in planetary conditions. Our original solid-solution calculations demonstrated a high solubility of carbon in solid hydrogen, and we are not aware of any metallic

system where the liquid solubility is lower than the solid.

To further address the referee’s concern, we have carried out extensive simulations to obtain the ensemble-averaged enthalpies of solution in the liquid, tabulated below. Unfortunately, the zero point energy contribution was calculated in the quasiharmonic approximation for solids, but this method cannot seriously be applied to liquids. calculate the ensemble-averaged enthalpic part of the solubility ΔH in the liquid, as shown in the table below. If we assume that the ZPE effects are similar to the solid, these enthalpies mean that the mixing is favoured, and more strongly so than in the solid. Thus we are confident in our assertion that carbon, oxygen and nitrogen will be fully soluble in metallic hydrogen and that, for planetary compositions, all these elements will dissolve.

Reference states	Enthalpy (eV/atom)	Compounds	Enthalpy (eV/atom)	ΔH (eV)
H ₁₂₈	-9.630(1)	CH ₆ in CH ₂₇₆	-10.116(1)	-0.16±0.38
C ₅₄	-144.087(2)	C ₂ H ₈ in C ₂ H ₅₇₆	-10.096(1)	-0.30±0.60
O ₄₈	-420.655(2)	C ₃ H ₁₀ in C ₃ H ₅₇₆	-10.326(1)	0.53±0.62
N ₆₄	-260.457(4)	CH ₄ OH in COH ₁₂₄	-14.007(1)	-5.99±0.15
		OH ₃ in OH ₁₂₄	-12.977(1)	-7.39±0.14
		NH ₄ in NH ₁₂₄	-11.665(1)	-3.60±0.15

Table shows the enthalpy change (ΔH) (eV/atom) of organic compounds in liquid metallic hydrogen, with reference states for carbon, oxygen, and nitrogen. For carbon at 500 GPa, the reference state is diamond. For oxygen at 500 GPa, investigations up to 250 GPa indicate that the monoclinic structure of the C2/m space group [Phys. Rev. B 76, 064101] is the most stable form from 100 to 250 GPa, so we use this structure as the reference state. For nitrogen at 500 GPa, it has been predicted [Phys. Rev. Lett. 102, 065501] that above 320 GPa, the helical tunnel structure of the $P2_12_12_1$ space group is the most stable. Our calculations imply a solubility of carbon in solid hydrogen, and we are not aware of any metallic system where the liquid solubility is lower than that in the solid phase. The table shows a high solubility of CO, O, and N in metallic hydrogen. These high ΔH values in CH₄OH, OH₃, and NH₄ could result from possible non-well-equilibrated ground-state references for oxygen and nitrogen.

2. In most of the paper, negative charge is claimed (but never demonstrated) for these molecules, but then on p. 9 the authors suddenly assign positive charges to CH₆, H₃O, NH₄. Which is correct?

We have done extensive further calculation to attempt to clarify the nature of the hypermolecules in term of covalent bonding. Please see the response to Reviewer 1, point 2 for details.

Using DFT to calculate charge on an object immersed in a charged background is difficult, because one has to determine whether the charge density “belongs” to the hypermolecule, an adjacent hydrogen, or to the free-electron background. To determine the nature of the hypermolecules alone, we carried out new DFT calculations with

the hypermolecule in a jellium background. These calculations show a free-electron density of electronic states with superposed sharp peaks corresponding to electrons localised on the hypermolecule. These peaks can be readily integrated to give two negative charges in CH_6 and, by way of checking, the expected neutral charge for CH_4 in jellium. The Friedel oscillations also suggest a screening of a charged object. Against that, the Mulliken charges on the atoms, and the occupation of the COPH orbitals indicate covalently bonded but neutral objects. The apparent contradiction can be traced to the hybridisation of the bonding states with free electrons states.

3. The explanation that added electrons enable more bonds is chemically unreasonable, as added electrons occupy in all these molecules antibonding orbitals and will lead to bond breaking.

Answer: We have carried out new COPH calculations using the LOBSTER code which enable us to relate our DFT calculation to a chemical viewpoint via extended Huckel theory. This shows that the antibonding orbitals, lie above the Fermi energy and are not occupied. Furthermore, it demonstrates that the valence electrons associated with the six CH bonds hypermethane can be assigned to C-H bonding orbitals, while other CH connections have no COPH bonding character.

4. Generally, increased coordination under pressure is normal, as in SiO_4 to SiO_6 change. Drawing chemical diagrams with 6 single C-H bonds is incorrect too. Here, bonds become fractional.

Answer: There are similarities and differences with SiO_6 : we mentioned it because silicon is in the same group as carbon and readers may be familiar with it. Also, the SiO_6 minerals such as perovskites and post-perovskites which we discuss have electrons donated from the cations. On the other hand, minerals with octahedrally coordinated Si are non-metallic, and the 2^- oxygens are "shared" between Si atoms forming a network, whereas the 1^- hydrogens form hypermolecules. On balance, we still think the analogy is close enough to be helpful.

What is very clear in the calculation, is that all six bonds are equivalent, hence we draw them as such. We believe that the combination of the jellium and COHP calculations demonstrates that there are six, chemically well-defined bonds. Whether they are fully occupied or fractional appears to depend on whether the extra two electrons are allocated to chemical bonds or the metallic band: i.e. on the choice of basis set .

5. Literature is cited sporadically (on silicates) and lacking recent research (on hydrocarbons under pressure).

Answer: We thank the referee for the suggestions. We have read a lot more literature and included more references to address this. There is a huge literature on hydrocarbons under pressure and so we concentrate here on work at extreme conditions of hundreds of GPa and in metallic environments, including shock wave, diamond anvil and simulations. It is entirely possible that there are other papers we haven't read or whose significance we missed, and we are happy to follow further suggestions.

Reviewer #3 (Remarks to the Author):

Reviewer #4 (Remarks to the Author):

The study uses density functional theory to characterize C-H bonding and molecule stability for conditions at which hydrogen is a metallic fluid; O-H and N-H bonding are also considered. Addressing chemistry at metallic-hydrogen conditions ought to be of interest to many disciplines, including chemistry, materials physics and planetary science.

The point of the paper is unclear, however, and it is likely to have minimal impact as currently written. First, it contains typographical errors, jargon and unclear material that make it hard for a nonspecialist to understand the assumptions, limitations and clearcut results of the work being presented. Fig. 3 is too small to read, and it is not obvious why 10 ps and 4x4x2 cells are adequate, for example. Second, the lack of hypotheses being tested or even proposed makes it hard for the reader to know what to make of this paper. The final section (Discussion and Conclusions) is notably confusing in acknowledging that no clear implications are being inferred for planets, nor is any experimental validation being proposed.

Answer: We are now explicit in the abstract that hypermolecules are a new type of chemistry, and comments of the previous referees show how novel and radical this suggestion is.

Regarding "lack of hypotheses", some background history is required. This project was initially to calculate accurately the solubility of carbon in fluid, metallic hydrogen. As a preliminary study, we looked at solid solution, and realised that the zero-point energy is overwhelmingly important, as is anharmonicity. For reasons mentioned by Referee 1, these energy differences cannot be taken from DFPT on a single snapshot in the fluid, which in any case misses the anharmonicity. So we resort to the MD-VACF. Unfortunately, this method doesn't work for fluids. However, by this point we noticed the far more interesting and serendipitous result that these hypermolecules were forming spontaneously in the simulation, so the focus of the paper switched. This historical perspective explains why it would be dishonest to claim that our hypothesis was that we were looking for the hypermolecules, which have emerged as the main result of the study.

We do still feel that the presentation order is correct, to start with the solids where we have more detailed and verifiable results, then move to the liquids which are of more general interest. However, we take the referees point that it would help the reader to include some "in this paper we..." paragraphs in the introduction to explain this, and thank the referee for the suggest.

Experimental verification is challenging with current methods. The synthesis of room temperature metallic hydrogen remains extremely controversial. There is a characteris-

tic phonon (Fig 1c) but detection will be difficult given that Raman and IR spectroscopic signals are almost non-existent for metallic systems. There should be indirect evidence, most notably the lack of a dense core in giant planets

In summary, the underlying work seems OK (though its reliability ought to be more thoroughly justified), but it is written up in a manner more suitable for a journal aimed at specialists rather than a general audience. It would help if the authors were to clarify whether they are describing a new kind of organic chemistry (the latter being defined as consisting of C-H-O-N compounds), or are demonstrating that C-H-O-N no longer form organic compounds at the conditions being considered (no longer characterized by the bonding and polymer formation of organic chemistry at ambient conditions). In either case, the significance of the results should be clearly stated.

We also improved the clarity of the discussion which, thanks to the suggestions of all referees, is now more strongly and clearly demonstrated in the paper.

Reviewer #5 (Remarks to the Author):

We thank the fifth referee for their efforts. Whichever co-review this was has led us to improve the presentation of our work.

Detailed response to referees

REVIEWER COMMENTS

Reviewer 2 (Remarks to the Author):

The manuscript has improved, although I am not fully satisfied with the authors's replies and still see major issues to be addressed (see below).

1. The calculations of enthalpic effects of incorporating impurities into pure hydrogen are not fully clear. Tables give as-calculated enthalpies, which are of no interest: the interesting quantity is the enthalpy of mixing which requires subtraction of enthalpies of pure compounds (pure H and pure C, and one should also try pure hydrocarbons stable at relevant conditions).

We have moved the tables presenting the as-calculated enthalpies to the supplementary material. The remaining table in the manuscript describes the enthalpy differences between the mixed compounds and the pure compounds.

2. The ratio C/H on Jupiter is much greater than 1 ppm and is in the range 0.1-1%. If 1 ppm of C is dissolved in hydrogen, then most carbon won't be in this form.

As far as we know, the overall composition of carbon in Jupiter is not known. The atmospheric C/H mass ratio is around 0.1% (C/H atomic ratio 0.01%) (Atreya et al). We can expect it to be higher in the core.

The solubility in solid hydrogen is very low, but in the case of the metallic fluid, which is relevant for Jupiter, our calculated enthalpy of solution is of order 100meV/atom which corresponds to an atomic solubility limit for carbon of almost 2% at 300K, and much higher at Jovian temperatures.

3. I am not sure what happened in the calculation, but strongly positive enthalpy of dissolution of C (0.39 eV) then changes into strongly negative values when discussing the incorporation of hydrocarbon molecules, and this is also problematic. The reason is that the formation of solutions is related to positive values of mixing energy (but if it is too large, solubility will be very small). Negative values of mixing enthalpy indicate the formation of compounds. The authors' conclusion of complete solubility (based on negative mixing enthalpy) is incorrect - what follows is that some amount of carbon will react and form some hydrocarbon, and that hydrocarbon at low T will form a separate solid phase and at high T may or may not mix with liquid hydrogen. This really needs to be investigated.

The strong positive enthalpy indicates that carbon is not soluble in the solid, whereas the negative enthalpy indicates it is soluble in the liquid. We agree that this is surprising, but we rechecked the calculation and it is correct. The reason seems to be that the hypermolecules can form in the liquid, but the I4/amd crystal symmetry makes them difficult to form and be incorporated in the solid without excessive distortions.

The extensive previous literature on "diamond rain" suggested, based on DFT, that carbon diamond coexists with metallic hydrogen. Even if other hyperhydrocarbons

exist, the important result is that diamond rain in metallic hydrogen is not plausible. Our work does not rule out that some other hydrocarbon compound is stable, and even forms a solid, but given the essential role of ZPE an exhaustive search would be a huge calculation. As with normal hydrocarbons, for example in crude oil, it is likely that many different compounds will coexist. Searching for more complex hypermolecules will be a topic for future work.

4. The formation of hypermolecules is indeed interesting, although in pure hydrocarbons this was not predicted, as far as I remember. Could this be a consequence of the hydrogen matrix?

Yes. These hypermolecules require the free electrons as well as the excess hydrogen atoms.

5. Conclusions of 6-valent carbon are toned down in the current manuscript, and should be removed altogether. Carbon has 4 valence electrons and cannot have valence greater than 4. To enable hexavalent state, you need to remove two electrons from carbon - then, C^{2+} has the ability to attract six electrons. Here it's very hard to see how this would happen. So one should only talk about increased coordination number. As a check, you can compare C-H distance in CH_6 with that in CH_4 at the same pressure (the distance in CH_4 should be shorter if my reasoning is correct).

We replaced "sixfold valence" with "sixfold coordination". One cannot compare CH_4 and CH_6 at the same pressure in H_2 because CH_4 immediately converts to CH_6 . We have done the comparison at equal electron density in jellium (SM table 9) where, as the referee expected, the CH_4 is indeed found to have significantly shorter bonds.

So the final conclusion is: major revision is needed.

Reviewer 3 (Remarks to the Author):

I have assessed the author response and changes in the manuscript. The authors have addressed with the previous concerns and produced additional data to support their findings. They have also made changes in the manuscript (specially adding a new section) and figures are much more visible. I think the manuscript is in better position than before, and I have no additional comments to the authors.

We thank the reviewer for the endorsement.

Reviewer 4 (Remarks to the Author):

The paper is much improved over the original. The additional work done in response to reviews helps, and the presentation has been improved with more legible figures and text. Although the authors have addressed Reviewer 1 and 2's more narrow, technical comments, the broader concerns remain to be addressed.

1) It is unclear what the authors mean by "organic compounds" and "organic-style molecules" (title and abstract). In response to review comments, they have now done a better job documenting covalent bonding (localized electron density between ions) and hybridization in their simulations (p. 9). It is unclear what these characteristics have to

do with organic chemistry as generally understood, for example with its rich diversity of polymeric structures.

Because it does not clarify, yet has the potential for causing misunderstanding, the authors should consider dropping the term “organic” in their title and text unless they can provide substantive reason for using it (they might consider using “covalent” instead). The hope is that publication of this work will not result in a press release leading people to think that metallic hydrogen might provide a habitat for life.

We really appreciated the referee comment “Because it does not clarify, yet has the potential for causing misunderstanding, the authors should consider...” Although it does not reflect on the calculations, it goes to the heart of good writing.

We have considered it, and we think our hypermolecules do meet the general definitions of organic compounds - although even wikipedia accepts this is not uniquely defined, e.g. https://en.wikipedia.org/wiki/Organic_compound

“Some chemical authorities define an organic compound as a chemical compound that contains a carbon–hydrogen or carbon–carbon bond; others consider an organic compound to be any chemical compound that contains carbon.”

To address the comment we now clarify and define precisely what we mean by organic in the introduction. We think this is the best term to describe molecules containing carbon and hydrogen, and/or oxygen, nitrogen, however, we do not want to introduce misunderstanding, so we are happy to take editorial guidance on this.

The evidence from our calculations does suggest that there may be “a rich diversity of polymeric structures”. Obviously redeveloping the entirety of organic chemistry in a free electron environment is well beyond the scope of one paper. We have presented examples of OH NH CH CC and CO bonds in a regime where HH bonds are unstable. We feel this is sufficient to demonstrate the idea. These calculations are already at the limit of our national supercomputer resources, and we can also reassure the referee that we have not tried, and failed to report, any hypermolecule which fell apart.

Our work provides no evidence for life, and we do not mention the word “life” anywhere in the paper.

2. The analogy between the structures of C- and Si-based oxides at high pressures has been documented through experiments, not just through computer simulations as in the present work. The authors ought to address the work of Yoo, Iota and colleagues on this matter [see Phys. Rev. B 82, 012105 (2010) and references 1-9 therein; see also the 2020 review at doi: 10.1063/1.5127897], with C-O compounds being more relevant to the issue than is the analogy with C-H compounds that they discuss.

The mention of sixfold carbon in CO₂ networks in the introduction is now extended to include references to the experimental work of Yoo’s group and the calculations of Scandolo.

As a detail, it is unclear what the authors mean by “pressure-unfavorable longer bondlengths” (p. 9, ¶ 3), as pressure tends to increase first-neighbor coordination numbers, which

increases bond lengths (first-neighbor distances) both in fluids and crystals.

We simply meant that at pressure, large molecular conformations will have a higher PV term in the free energy. The referee is right that it is not unusual for pressure to induce more efficient packing at the expense of increasing molecular size. We could not detect Jahn-Teller distortions in our high temperature MD. In our jellium calculations there are no neighbours.

In view of the referee's possible counterexample, we removed this speculative clause.

3. The authors' conclusions (e.g., p. 4 near bottom, p. 9 near top) of various elements being soluble at the $10^0 - 10^2$ ppm level in metallic hydrogen (i.e., for Solar/Jovian abundances of all elements other than He) is hardly surprising or useful, as it is difficult to avoid this level of solubility due to the entropy of mixing in any fluid at planetary interior temperatures.

The reviewer is quite correct that these are unsurprising solubilities (although He is predicted to almost fully demix from metallic hydrogen, even at 5000K). The point we are trying to make is not that 100 ppm is a high solubility, rather that it accounts for all the carbon in the expected composition of these planets.

4. Perhaps most disappointing in the response to reviews and in the revised text is that the authors indicate no way of empirically validating or falsifying their results, whether through experiments or observations on planets.

Experiment has always lagged theory in studies of metallic hydrogen, and we did not wish to claim experiments could be done when they currently cannot.

The original 1935 prediction of metallic hydrogen stated that "the pressure necessary for the transformation is 250,000 atmos., which is outside the scope of the present technique.". At that stage Bridgeman had reached around 100,000 atmos. and the 50-fold increase required to reach 500GPa seemed fanciful. But nowadays many labs can do this, although whether metallic hydrogen has actually been made remains controversial.

So the PT conditions considered here are currently achievable, although metallic hydrogen has technical challenges from corrosion and diffusion which make it problematic. It is well known that diamond anvil cells require special coatings to work with high pressure hydrogen, and easiest probe for molecular vibrations, Raman spectroscopy, provides poor signal from a metal. This is not our speciality, but these seem like problems which could be solved in the next few years.

Reviewer 5 (Remarks to the Author):

We thank the ECR reviewer for their efforts.

Reviewer 6 (Remarks to the Author):

I have carefully read the revised manuscript and the reports of reviewers no. 2 and no. 4 on the latest version. I share the scepticism expressed by both reviewers. On the other hand, I think the topic is interesting enough for publication in Nature Communications. The authors should be given another chance to make changes and/or add to the paper before a final decision is made.

We thank the reviewer for the endorsement.

Reviewer 7 (Remarks to the Author):

This Referee will be only concerned about the following issues:

1.) Is 6-coordination of carbon atoms something we do not know in organic chemistry? The clear answer is no. Non-classical carbo cations is the keyword. The recent publication (mali reports on covalently six-coordinated carbon species in pentagonal pyramidal molecules $[C_6(CH_3)_6]^{2+}$, where each C atom has an exo ligand CH₃; 5 C atoms form a ring capped by the 6th carbon atom. This stable cluster is equivalent to a nido borane B₆H₁₀ and well understood from the Lipscomb's work (who synthesized the cluster in 1954, determined its crystal structure 1958, and explained its bonding pattern, see also his noble lecture (Fig. 14 in Science 1977, 196, 1047ff) and the Wade's rules (1971).

We believed that our work was the first prediction of six-coordinated covalent carbon. We were wrong, it has in fact already been experiment synthesized in a molecular ion. So this result, which Referee 2 found controversial, is in fact already empirically known.

We thank the referee for bringing this literature to our attention. We removed the claim that our sixfold covalent bonding was "new". We mention Wade's rules and their extension to carbon by Jemmis. Referee 4 also mentioned a 2+ hexavalent state.

2.) Do the ELF plots provide evidence of covalent bonds between central C and neighbor H atoms in CH₆@H(metal)? The clear answer is no. The 'smoking gun' feature of a covalent chemical bond in the framework of ELF analysis of chemical bonding is the creation of a local maximum (attractor) between the atoms involved. This is not seen in the ELF diagrams shown (please avoid calling them 'ELFs'; for the same reason you were not writing that you were doing a couple of 'DFTs' to investigate the energetic situation of C in metallic H). The attractors seen there are at the position of H nuclei, which is unfortunately something you always see on H in any compound. Since ELF shows shell structure, it gives a maximum for the 1st shell of H, which accidentally is also the valence region. This way also in classical ethane C₂H₆ the C-H bond does only show a maximum at the H position related to the shell structure already, while the C-C bond shows a new attractor between the carbon atoms. This is the ELF proof a C-C bond here. The high values between C and H seen in the ELF diagrams Fig. 1 and SI Fig. 6 of the ms. do not indicate a special type of covalent bonds; if you look into literature (Savin, Nesper, Wengert, Faessler, ELF: The Electron Localization Function, Angew. Chem. Int. Ed. 1997, 36, 1808) you see in Fig. 2a a plot of ELF for

ethane with the C-C bond attractor, and also in Fig. 2c the ELF distribution between two non-bonded H atoms at the same C atom in C₂H₄; the values are quite high there like in the diagrams shown for CH₆@H(metal).

The ELF does not provide "smoking gun" evidence for the covalent bond because there is no interatomic ELF maximum. But, as with the ethane example, where it is agreed the CH is a covalent bond, the absence of such a maximum does not rule out covalency. However, the high value of ELF does indicate that there are localised electrons: the bonding is not metallic.

We don't have any disagreement with the referee here, the ELF looks like ELF often does for CH bonds, it's high value rules out metallic bonding, but ELF is a less convincing indicator of covalency than the COHP.

A remark on the ELF diagrams shown: The core (1s) region of C has been omitted (why?). It would be interesting to see this region for a compressed situation with such a high pressure. Does C(1s) density leak out into the valence regions.

The 1s electrons in carbon are very low lying (around 500 eV) and compact (around 0.1Å) so are included in the pseudopotential, as is standard for this type of calculation. They do not hybridise or leak out into the valence region. Consequently, the localised 1s state is not evaluated in the ELF. We agree that this could be confusing, so we added a clarifying note in the caption.

3.) Concerning section 2.4 Chemistry of hypermolecules: The decisive sentence seems to me "... analysis whether the bonds are fractionally occupied is complicated because the bonding states are degenerate with free electrons." This clearly indicates that the 'molecules' are not at all isolated (like CH₄ in our atmosphere of N₂, and O₂) but embedded in the 'H matrix' like an arbitrary fragment of an alloy is. I would tend to believe that COHP results indicate that C is covalently bonded to 6 H atoms of this matrix. However, and this has not been investigated, I guess the ligating H atoms would be found to be still bonded to the matrix but slightly less than the undisturbed H atoms of the matrix. Thus, there are no separate CH₆ molecules in this matrix.

We carried out the additional calculations as requested by the referee. We see that the CH bond has a significantly larger bonding character than HH connections to the second shell, HH connections far from the carbon, and HH connections in I₄/amd crystal hydrogen. This is further support for describing the CH₆ as covalently bonded, clearly distinct from the fourfold-coordinated hydrogen network in I₄₁/amd which would not be described as "covalent" or "tetravalent" hydrogen.

Figures show the COHP interactions in panels (a) and (b). Panel (a) presents the COHP interactions between the hydrogen atoms in the first shell, as illustrated in Figure (c). Panel (b) shows the COHP interactions between hydrogen atoms in the first and second shells. In panel (b), we also compute the COHP interaction between nearby hydrogen atoms in the solid metallic background of CH₁₂₄, which may be zero for some pairs, as well as the COHP interaction between hydrogen atoms in solid $I4_1/amd$ metallic hydrogen, using a conventional cell with four hydrogen atoms.

To my opinion it does not help nowadays going to simpler models like the jellium one

if we have not understood it with first-principle methods like DFT.

We believe that the jellium calculation gives important insight into the system. In particular, it demonstrates that these hypermolecules are stabilised by the high electron density, and not the pressure *per se*. In practice, one has to go to high pressure to achieve high electron density, but we think the distinction is worth making.

Conclusion: I really like the study and the computational work, but the bonding discussion is not at the same high level as the remaining part. I cannot see molecules like CH₆ as separated entities in the metallic H matrix. Moreover, six-coordinated C atoms are known in organic chemistry since the second half of the 20th century, which may destroy the quest of hype.

The jellium calculation demonstrates that CH₆ is a stable entity in a sufficiently high electron density: no surrounding ligand shell is required. We were not aware of Malichevski's experimental discovery of the 6-coordinated carbon. We added references and brief discussion to give due credit to that work.

Dear Editor,

We have addressed all remaining reviewers' comments, as shown below this letter.

We revised the title of our work, and revised the abstract and the manuscript text accordingly. We discussed more about JUICE mission as the on-board instruments might be able to reveal the value of the tidal Q of Jupiter, and would link to the structure of its core.

We have completed all the checklist items. They are overwhelming. If you have further inquiries, please do not hesitate to get back to us.

We hope that our manuscript can be considered as a regular article in *Nature Communications*.

Sincerely Yours,
J.Seeyangnok, U.Pinsook, and G.J.Ackland

EDITORS' COMMENTS

First, we ask you to revise your paper to address our editorial requests (in the attached Author Checklist) and any remaining comments from reviewers (included at the end of this email, if applicable).

We have edited the manuscript according to editorial requests as listed in the attached Author Checklist. We sincerely thank the editor for the valuable comments and suggestions.

REVIEWERS' COMMENTS

Reviewer #2 (Remarks to the Author): I am satisfied with the revised manuscript and suggest its acceptance

We would like to express our sincere gratitude to Referee #2 for the careful reading of our manuscript and the thoughtful comments and suggestions provided. Your constructive feedback greatly helped us clarify several key points and improve the overall quality of the work.

Reviewer #5 (Remarks to the Author): I co-reviewed this manuscript with one of the reviewers who provided the listed reports. This is part of the Nature Communications initiative to facilitate training in peer review and to provide appropriate recognition for Early Career Researchers who co-review manuscripts.

We sincerely thank Referee #5 for the valuable time and effort spent in reviewing our manuscript. Your detailed observations and insightful recommendations have been very helpful in refining both the content and presentation of our study.

Reviewer #7 (Remarks to the Author): The revised manuscript has considered my remarks in a sufficient way. I do not want to open further discussion about chemical

bonding in these compounds because the authors have been done enough to substantiate their ideas. Of course, this will stimulate discussion and further investigations using different techniques, which is even an effect implicitly or explicitly intended by such a study.

The only thing that disturbs me as a chemist is the classification of H₂O and NH₃ as organic molecules. Of course, the molecules cannot defend themselves against this (i.e., it is not a measurable property), but a look into organic and inorganic chemistry textbooks (at least those that I know) will show that these molecules are treated in the inorganic chemistry ones.

The title of the study may be interpreted as being related to only a part of the molecules actually computed, which I feel is ok. However, in the abstract "organic-style molecules OH₃, NH₄" are explicitly mentioned, which is unconventional at least. It would not harm the interest in and the significance of the results if these two molecules were more conventionally classified. It will even show that not only saturated "organic molecules" become hydrogenated but also inorganic ones. The chemically important and new issue is that saturated molecules like CH₄ and C₂H₆ (see below) get hydrogenated, while it is common knowledge (freshmen courses of organic chemistry) that this is possible only for unsaturated hydrocarbons, e.g. $2\text{HC}=\text{CH}_2 + \text{H}_2 \rightarrow 3\text{HC}-\text{CH}_3$.

Summarizing, I have nothing more to complain, just a few friendly recommendations left free to follow for the authors.

We are grateful to Referee #7 for the thorough and critical review of our manuscript. Your comments highlighted important aspects that required clarification, and your suggestions contributed meaningfully to the improvement of our work.